# Structural basis of adenylyl cyclase 9 activation

Chao Qi [1,2], Pia Lavriha[2], Ved Mehta [1,2], Basavraj Khanppnavar[1,2], Inayathulla Mohammed[3], Yong Li[4], Michalis Lazaratos[5], Jonas V. Schaefer[6], Birgit Dreier[6], Andreas Plückthun [6], Ana-Nicoleta Bondar[5,7], Carmen W. Dessauer[4] & Volodymyr M. Korkhov [1,2✉]

Adenylyl cyclase 9 (AC9) is a membrane-bound enzyme that converts ATP into cAMP. The enzyme is weakly activated by forskolin, fully activated by the G protein Gαs subunit and is autoinhibited by the AC9 C-terminus. Although our recent structural studies of the AC9-Gαs complex provided the framework for understanding AC9 autoinhibition, the conformational changes that AC9 undergoes in response to activator binding remains poorly understood. Here, we present the cryo-EM structures of AC9 in several distinct states: (i) AC9 bound to a nucleotide inhibitor MANT-GTP, (ii) bound to an artificial activator (DARPin C4) and MANT-GTP, (iii) bound to DARPin C4 and a nucleotide analogue ATPαS, (iv) bound to Gαs and MANT-GTP. The artificial activator DARPin C4 partially activates AC9 by binding at a site that overlaps with the Gαs binding site. Together with the previously observed occluded and forskolin-bound conformations, structural comparisons of AC9 in the four conformations described here show that secondary structure rearrangements in the region surrounding the forskolin binding site are essential for AC9 activation.

[1] Institute of Molecular Biology and Biophysics, ETH, Zurich, Switzerland. [2] Laboratory of Biomolecular Research, Division of Biology and Chemistry, Paul Scherrer Institute, Villigen, Switzerland. [3] Biozentrum, University of Basel, Basel, Switzerland. [4] Department of Integrative Biology and Pharmacology, McGovern Medical School, University of Texas Health Science Center, Houston, TX, USA. [5] Department of Physics, Theoretical Molecular Biophysics Group, Freie Universität Berlin, Berlin, Germany. [6] Department of Biochemistry, University of Zurich, Zurich, Switzerland. [7] Present address: University of Bucharest, Faculty of Physics, Str. Atomiştilor 405, Bucharest-Măgurele 077125, Romania Institute for Neuroscience and Medicine and Institute for Advanced Simulations (IAS-5/INM-9), Computational Biomedicine, Forschungszentrum Jülich, Jülich 52425, Germany. ✉email: volodymyr.korkhov@psi.ch

Adenylyl cyclases (ACs) play a fundamental role in many G protein-coupled receptor (GPCR) mediated signal transduction pathways[1–3]. Activation of a Gαs-coupled GPCR by an extracellular stimulus, such as a hormone, leads to a cascade of events that include the exchange of GDP bound to the Gαs subunit to GTP, dissociation of the GTP-bound Gαs from the heterotrimeric G protein-GPCR complex, followed by binding of the GTP-bound Gαs to a membrane AC. The interaction with Gαs potentiates the ACs ability to convert a molecule of adenosine 5'-triphosphate (ATP) into cyclic adenosine monophosphate (cAMP)[1,4]. The produced cAMP is a key second messenger in many living cells, binding to and regulating a number of downstream effector proteins, and thus modulating a plethora of physiological functions[5]. The nine subtypes of membrane ACs described in mammals, AC1-9, differ in cellular localization, tissue distribution, and physiological functions[5]. For example, the ACs predominantly expressed in the nervous system, AC1 and AC8, are linked to cognitive processes and pain perception[2], whereas in heart, AC5, AC6, and AC9 are linked to heart disease[2,6]. Mutations of several membrane ACs have been linked to genetic diseases, including autosomal deafness 44 (AC1)[7], obesity and type 2 diabetes (AC3)[8], familial dyskinesia with facial myokymia (AC5)[9], or lethal congenital contracture syndrome 8 (AC6)[10].

The membrane ACs share a conserved predicted domain arrangement, membrane topology and a high degree of sequence similarity, particularly in the catalytic regions of the protein[11]. Each membrane AC contains two conserved cytosolic catalytic domains and twelve predicted transmembrane (TM) helices, with cytosolic N- and C-termini of varied lengths and regulatory roles[1,11]. Insights into the structure and molecular mechanism of the membrane ACs have been gained by the early X-ray crystallographic studies on the chimeric soluble domain of adenylyl cyclase (AC5$_{c1}$/AC2$_{c2}$) in complex with Gαs and forskolin[12–14]. These studies provided the structural basis for the two metal-ion-catalysis of ATP-cAMP conversion, revealed some of the intermediate states of the enzyme, and provided a plausible explanation of enzyme activation by the plant-derived small molecule activator, forskolin[15]. While forskolin is a non-physiological activator of the membrane ACs, an endogenous molecule that regulates the ACs at the allosteric binding site has not yet been identified. Forskolin and Gαs were required to stabilize the dimer of the isolated AC5$_{C1}$ and AC2$_{C2}$ domains, as both regulators enhance the affinity between C1a and C2a in the soluble system[12–14]. The mechanism of forskolin-mediated AC activation was ascribed to its ability to "lock" the two domains closer together, likely inducing a conformational change to reorient C1a and C2a with respect to each other to enhance activation[12–14]. Recently we determined the cryo-EM structure of the full-length bovine AC9 bound to Gαs in an autoinhibited "occluded" state, together with the structure of a C-terminally truncated AC9 (AC9$_{1250}$), bound to Gαs, forskolin and MANT-GTP, a non-cyclizable AC inhibitor[1]. These structures provided the first glimpse into the architecture and auto-inhibitory regulation of a full-length membrane AC. Furthermore, the ability of forskolin to activate AC9 has been a controversial subject, with some studies indicating that the enzyme is insensitive to forskolin[2], some studies showing that mouse AC9 could be converted to forskolin sensitive cyclase by a Tyr1082Leu mutation[16]. Recently, the cryo-EM structure of AC9$_{1250}$–Gαs bound to MANT-GTP and forskolin, combined with biochemical studies, confirmed that AC9 can be activated by forskolin binding to its canonical allosteric site in the presence of Gαs[1,17], further suggesting that AC9 is not completely insensitive to forskolin.

AC9 is expressed ubiquitously and plays an important physiological role. A polymorphism of AC9, rs2230739, has been associated with differences in susceptibility to drug treatment in asthma patients, indicating AC9 as a potential asthma drug target[2,18]. In the heart, AC9 is involved in the slow delayed rectifier (I$_{Ks}$) current through its interaction with the A-kinase anchoring protein, Yotiao and KCNQ1/KCNE1 channels[19]. AC9 has also been suggested to have a cardioprotective role in the heart through its interaction with Hsp20[6].

Despite the availability of the high-resolution X-ray structures of the AC5$_{c1}$/AC2$_{c2}$ domains as well as the single particle cryo-EM structures of AC9, the molecular determinants of AC9 regulation remain unclear. We set out to determine the key conformational changes associated with distinct functional states of AC9. The choice of the AC9 as a model was motivated by both its tremendous biomedical significance and relevance to disease, as well as by our ability to probe the structure of the full-length protein. To aid our structural and biochemical analysis we developed an artificial binder of AC9, a designed ankyrin repeat protein (DARPin)[20] that binds to AC9 and partially activates it.

Here, we present four cryo-EM structures: (i) AC9 in complex with MANT-GTP (AC9–M), (ii) AC9 in complex with MANT-GTP and an artificial partial activator, DARPin C4, which is capable of activating the AC9 without inducing the occluded state (AC9–C4–M), (iii) a DARPin C4-bound state with ATPαS instead of MANT-GTP (AC9–C4–A), (iv) AC9$_{1250}$ in complex with Gαs and MANT-GTP, in the absence of forskolin (AC9$_{1250}$–Gαs–M). Analysis of all structures of AC9 available to date reveals the conformational transitions that the protein goes through in response to binding of distinct activating agents (DARPin C4, Gαs, forskolin). The magnitude of the conformational changes correlates with the potency of the activator. Our data suggest that activation of AC9 is intricately linked to the conformation of the allosteric site.

## Results

**Structure of AC9 bound to MANT-GTP.** To characterize the conformation of AC9 in the absence of G protein α subunit, we determined the structure of bovine AC9 in complex with MANT-GTP (an inhibitor we used to stabilize AC9), referred to as AC9–M throughout the text below, using cryo-EM and single-particle analysis at 4.9 Å resolution (Supplementary Fig. 1). MANT-GTP is not a substrate analog of AC9, and way it is accommodated in the active site of the ACs is known to differ from that for ATP analogs. Our choice of MANT-GTP as a ligand for the active site of AC9 was based on two considerations: (i) we have used it successfully in our previous studies of membrane ACs, (ii) the overall conformation of the AC catalytic domains stabilized by MANT-GTP is similar to that stabilized by ATP analogs, such as ATPαS, based on X-ray crystallographic studies[13,21]. Due to the low molecular weight and innate flexibility and pseudo-two-fold symmetry of the full-length AC9, the resolution of the AC9–M reconstruction could not be improved beyond 4.9 Å. Nevertheless, the structure revealed the main features of the protein, including the TM1-12 helices, the helical domain, and the catalytic domain. The positions of these domains could be assigned to the density features unequivocally (Supplementary Fig. 1).

**Artificial partial activator of AC9, DAPRin C4.** In order to stabilize the protein, increase its size for cryo-EM analysis and potentially break the pseudo-two-fold symmetry, we selected a panel of designed ankyrin repeat proteins (DARPins) by using ribosome display[20,22,23]. From a small pool of 21 DARPins that specifically bound to AC9, we identified DARPin C4, capable of activating AC9 alone, but not the AC9–Gαs protein complex (Fig. 1a, b). Analysis of the AC specificity of DARPin C4 revealed

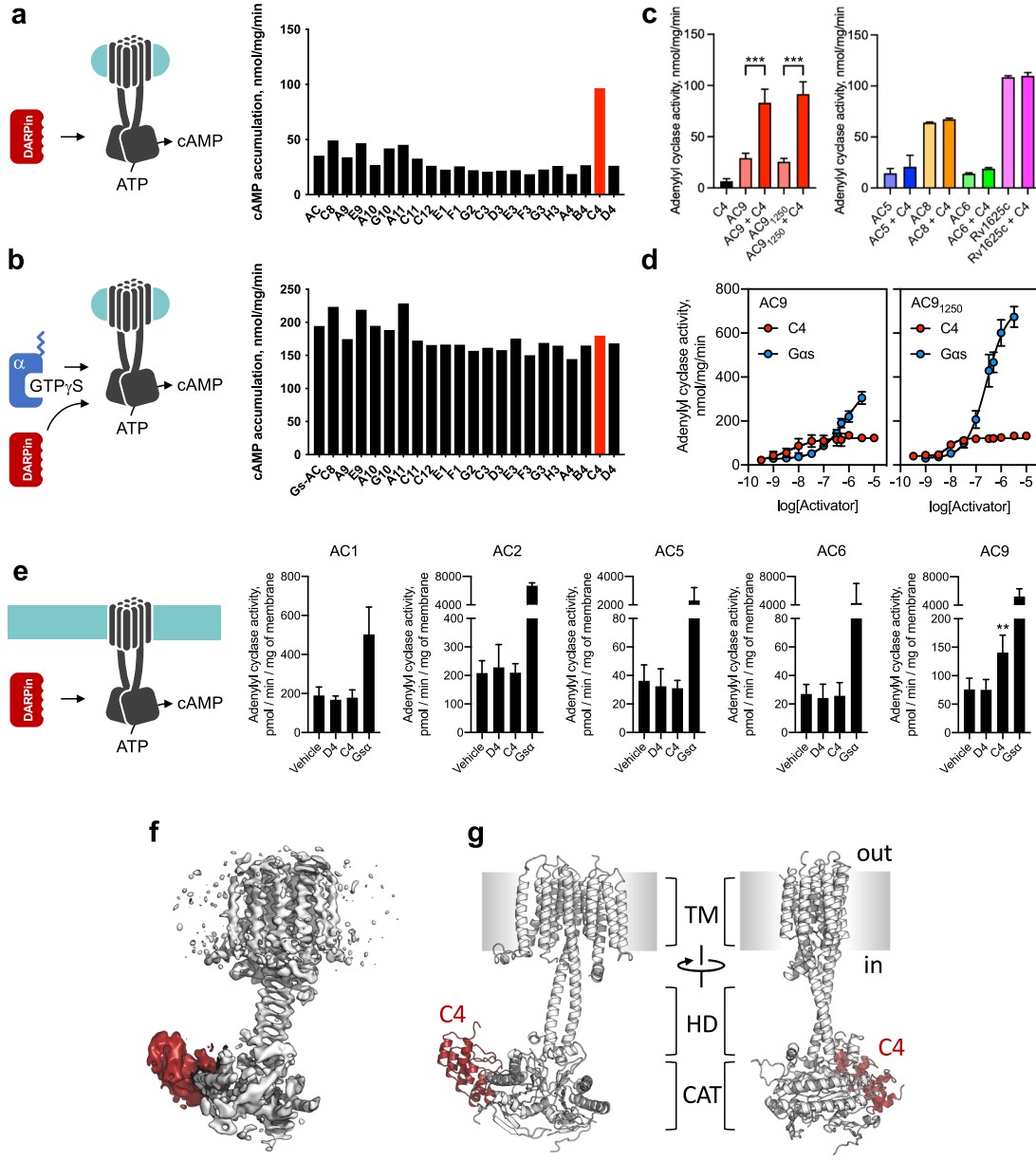

**Fig. 1 Identification, biochemical properties and structure of DARPin C4 as a partial activator of AC9. a–b** AC9 activity screens identify the DARPin C4 as an AC9 activator; the experiment in (**b**) was performed in the presence of GTPγS-bound Gαs (data are shown as mean values, $n = 2$). Here and in all other panels $n$ refers to independent experiments. Source data are provided as a Source Data file. **c**, Activity assays of purified, detergent-solubilized ACs (as in (**a**, **b**)) demonstrate selectivity of DARPin C4 for AC9 (data are shown as mean ± SEM, $n = 3$ for all datasets; $n = 4$ for AC9 and AC9$_{1250}$). The values were compared using one-way analysis of variance (ANOVA), followed by Tukey's multiple comparisons test; for AC9 vs AC9 + C4, $P = 0.0002$ (***), for AC91250 vs AC9$_{1250}$ + C4, $P < 0.0001$ (***). $P$ values for all comparisons are available in the Source Data file. **d** Dose–response activity curves of the DARPin C4 show that it has a high apparent affinity for AC9 (left) and for AC9$_{1250}$ (right); unlike Gαs, DARPin C4 acts as a partial activator of AC9 regardless of the presence of the C2b domain (data are shown as mean ± SEM, $n = 3$; for AC9$_{1250}$ + C4, $n = 4$). **e** AC activity assays performed using membrane preparations of Sf9 cells expressing different ACs confirm that activation by DARPin C4 is selective for AC9 (data are shown as mean ± SEM, $n = 4$). The values were compared using one-way analysis of variance (ANOVA), followed by Tukey's multiple comparisons. For AC9 vehicle vs C4, $P = 0.0091$ (**); for vehicle vs D4, $P = 0.9986$; Gαs data were excluded in analysis). **f** Cryo-EM map of the DARPin C4-bound AC9 (white map corresponds to AC9, red—DARPin C4. **g** Model of the AC9–C4 complex; key elements of the structure, including the transmembrane domain bundle (TM), the helical domain (HD), the catalytic domain (CAT) and the DARPin C4 (C4, red) are indicated.

that it is highly selective for AC9 (Fig. 1c, e). The affinity of the DAPRin C4 for purified AC9 in detergent micelles, based on AC activity assays, is in the high nanomolar range (Fig. 1d). The cAMP accumulation assays showed a half-maximal effective concentration (EC$_{50}$) for DARPin C4 activation of AC9 of 4.4 nM (8.3 nM for AC9$_{1250}$), with very similar $V_{max}$ values (~120 nmol/ mg/min for both AC9 and bAC9$_{1250}$). These $V_{max}$ values are ~3-

fold higher compared to the basal activity of AC9 (~37 nmol/mg/ min)[1]. In comparison, Gαs potently activated the full-length AC9 with a ~15-fold increase for the full-length protein and a ~24-fold increase for AC9$_{1250}$ (i.e., in the absence of the C2b domain) (Fig. 1d). Forskolin dose–response assay showed that DARPin C4-bound AC9 was sensitive to forskolin, with EC$_{50}$ about 130 μM, similar to that of Gαs-AC9 (~110 μM)[1]. Experiments using

membrane preparations of insect cells expressing select AC isoforms confirmed the selectivity of DARPin C4 for AC9 (Fig. 1e) and the high affinity of this agent for AC9 under a variety of experimental conditions (i.e., in the presence of $Mn^{2+}$ or $Mg^{2+}$ ions that differentially support the activity of AC9; Supplementary Fig. 2).

**In vivo effects of DARPin C4.** Expression of AC9 in mammalian cells was not influenced by co-expression of DARPin C4 (Supplementary Fig. 3). To determine whether DARPin C4 can interact with AC9 in a cellular context, we used FRET microscopy. Analysis of FRET between the YFP-tagged AC9 and CFP-tagged DARPin C4 confirmed the interaction between these proteins (Supplementary Fig. 4). Furthermore, expression of DARPin C4 in HEK293F cells led to increased cAMP accumulation when co-expressed with AC9, compared to control cells (Supplementary Fig. 4). These proof-of-concept experiments established unequivocally that DARPin C4 interacts with and activates AC9 in cells, and thus can be used in cell-based applications.

**Structure of AC9 bound to DARPin C4.** To elucidate the structural basis of AC9 activation by DARPin C4, we copurified these two proteins using size exclusion chromatography (SEC) (Supplementary Fig. 5) and subjected the AC9–C4 complex to cryo-EM imaging. Using single-particle analysis of the complex we obtained the 3D reconstruction of AC9–C4–M (AC9–C4 complex in the presence of 0.5 mM MANT-GTP) at a resolution of 4.2 Å (Fig. 1f, g, Supplementary Figs. 6, 7). The structure revealed all the previously observed features of the AC9 (Fig. 1g), including the 12-TM domain region (TM), the helical domain (HD), and the catalytic domain (CAT). The DARPin C4 binds to the C2a domain of AC9 at a site that partially overlaps with the Gαs-binding site (Figs. 1f, g, 2a–g). The variable region of DARPin C4 inserts into the groove formed by the α2′ and α3′ helices of the C2a domain of AC9 (Fig. 2a, f), similarly to the way the switch II region of Gαs binds to and activates the enzyme[1].

**Determinants of AC9 activation by DARPin C4.** Both activator proteins, Gαs and DARP C4 bind to a very similar region of the C2a domain (Fig. 2d, e), yet their effects on the functional state of the protein are drastically different. There is one major structural difference between the structure of AC9–C4–M and that of AC9–Gαs determined previously[1]: DARPin C4 does not appear to stablise the occluded state, in which the C2b domain of AC9 displaces the nucleotide and forskolin from their binding sites at the C1a/C2a domain interface. In contrast, Gαs induced the I1263-P1275 region of the C2b domain to wedge itself into the substrate and allosteric binding sites of AC9, presumably as part of an auto-inhibitory mechanism that prevents over-production of cAMP[1,24].

The remarkable difference between the ability of the DARPin C4 and Gαs protein to activate AC9 and to induce the auto-inhibitory occluded conformation shows that there is a fundamental difference between the molecular interactions of these two proteins. Comparisons of the binding interfaces between AC9 and its regulators, Gαs and DARPin C4, provide clues to the mechanism of AC9 activation by the DARPin C4. A subset of shared residues between the G protein and the DARPin C4-binding sites in the C2a domain, unique to AC9, is likely responsible for the ability of the DARPin C4 to specifically recognize this enzyme (Supplementary Fig. 8a–c). Furthermore, a loop in the C1a domain corresponding to residues I380-P384, appears to be flexible and could not be resolved in the AC9–C4–M map (Fig. 2f). However, this loop interacts with the Gαs protein, evident from the structure of the AC9–Gαs complex (Fig. 2g)

as well as the available $AC5_{c1}/AC2_{c2}$ crystal structures. Gαs as a full activator engages C1a and C2a domain simultaneously, whereas DARPin C4 as a partial activator only engages the C2a domain. This extensive interaction with the regions in both C1a and C2a domain of AC9 may contribute to higher potency of Gαs, compared to DARPin C4.

The structure of AC9–C4–M revealed density in the active site consistent with the presence of MANT-GTP, as well as a lack of any density in the allosteric site (Fig. 3a). Although the AC9–C4 structure provides clues about elements of AC9 participating in selective binding of DARPin C4, it is unclear why the occluded state of AC9 has only been observed in the presence of Gαs. Moreover, we could only obtain an activated state of the protein bound to Gαs upon removal of the C-terminal C2b domain. In contrast, in the case of the DARPin C4-bound state, the full-length AC9 could be captured in a nucleotide-bound state. It is apparent that there is an intricate link between the Gαs protein and the largely unstructured C-terminus of AC9, which will require further examination.

**AC9-bound nucleotide conformation in the absence of forskolin.** Although the cryo-EM maps of AC9–M and AC9–C4–M featured the density elements corresponding to a bound MANT-GTP molecule (Fig. 3a), the location of the nucleotide density in each of the maps was distinct from that observed in the previously determined $AC9_{1250}$–Gαs structure in the presence of MANT-GTP and forskolin ($AC9_{1250}$–Gαs–MF) (Fig. 3f)[1]. The nucleotide density stretched out towards the forskolin binding site, suggesting that the molecule may have been captured in a previously undescribed orientation within the active site (Fig. 3a). The local resolution of this region in both density maps (AC9–M and AC9–C4–M) was not sufficiently high to unambiguously define the pose of MANT-GTP.

To further structurally characterize the nucleotide-binding site of AC9, we determined the structure of $AC9_{1250}$–Gαs–M (the truncated $AC9_{1250}$ in complex with Gαs and MANT-GTP). Focused refinement of the 3D reconstruction improved the resolution of the map corresponding to the soluble part of the complex to 3.8 Å (Supplementary Figs. 9, 10). As in the case of AC9–M and AC9–C4–M, the $AC9_{1250}$–Gαs–M map featured the density of the nucleotide extending towards the allosteric site (Fig. 3a, e). The previously determined nucleotide pose "M1" corresponds to the structures of $AC9_{1250}$–Gαs–MF[1] (Fig. 3f), the $AC5_{c1}/AC2_{c2}$–Gαs[21] and the *M. intracellulare* Cya[25] (Supplementary Fig. 11a, b). The AC catalytic site is known to be capable of accommodating the nucleotides in non-canonical orientations, evident from two distinct poses of the MANT-GTP in the mycobacterial Cya[25] (Supplementary Fig. 11b), the structure of sAC bound to ApCpp (also known as AMP-Cpp) and an inhibitor LRE1[26] (Supplementary Fig. 11d), or the structure of *M. tuberculosis* MA1120 bound to $Ca^{2+}$ and ATP[27] (Supplementary Fig. 11e). The density occupying the nucleotide-binding site observed in AC9–C4–M and $AC9_{1250}$–Gαs–M maps is consistent with a previously undescribed non-canonical pose of MANT-GTP, featuring the MANT-group stretching out towards the forskolin binding site.

This unexpected observation prompted us to further investigate nucleotide binding to AC9. Although MANT-GTP is a nucleotide-derived inhibitor, its structure is quite distinct from the natural AC substrate ATP. To better understand the observed extended density protruding from the nucleotide-binding site, we determined the structure of AC9 in a complex with DARPin C4 and an ATP analog ATPαS (AC9–C4–A) at 4.3 Å resolution (Supplementary Fig. 12, 13). While ATPαS is also an AC inhibitor, it lacks the MANT-group and is likely a better representative of the nucleotide pose in the ATP-bound state of AC9. The structure of AC9–C4–A is very

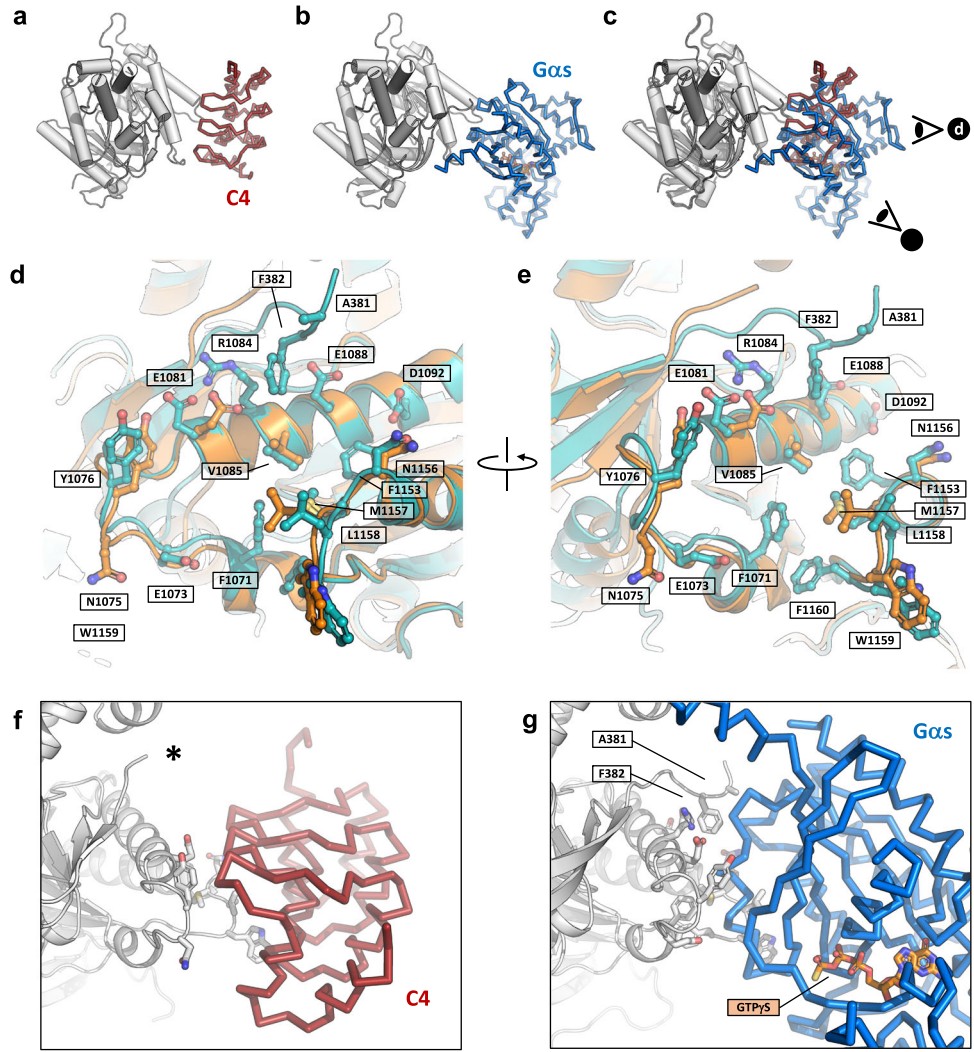

**Fig. 2 DARPin C4-binding site within the catalytic domain of AC9. a–c** Views of the DARPin-C4- (**a**) and Gαs-bound AC9 (**b**). **c** The binding sites of the two interaction partners of AC9 overlap. The symbols in (**c**) indicate the points of view in (**d**) and (**e**). **d–e** The views of the activator binding sites in the DARPin C4- (orange) or Gαs-bound (blue) AC9 complexes. The side-chain of all residues within 4 Å of the interaction partner are shown as sticks. **f–g** The views of the AC9–activator interface, revealing the absence (**f**, indicated by an asterisk) and the presence (**g**) of interaction between AC9 and DARPin C4 and Gαs, respectively, involving the C1a domain loop region (residues A381 and F382 in proximity with G protein are indicated).

similar to AC9–C4–M, with the full model and soluble domain RMSD of ~2 Å and ~1 Å, respectively. The absence of the stretched density observed in all of the forskolin-free MANT-GTP-bound structures (Fig. 3b) indicates that the stretched density in AC9–C4–M and AC9$_{1250}$–Gαs–M map corresponds to the MANT-group. We propose that MANT-GTP in the absence of forskolin adopts a stable MANT-GTP pose "M2", which is distinct from the M1 pose and is characterized by the MANT-group extending towards the forskolin site. This MANT-GTP pose is consistent with our previous observations of forskolin decreasing the IC$_{50}$ of MANT-GTP[1]. The allosteric coupling between the two compounds is manifested by MANT-GTP moving from pose M2 to M1 upon forskolin binding.

Although ATP binding and catalysis by ACs have been extensively investigated previously, the unusual binding pose of MANT-GTP observed in our structures raised the question whether an ATP molecule could also be accommodated in an alternative binding pose in the active site. To determine whether AC9 may have any preference for a specific nucleotide pose, we performed molecular dynamics (MD) simulations using the cytosolic portion of AC9–Gαs model. We substituted the MANT-GTP molecules in M1 and M2 poses with the molecules of ATP

placed in the corresponding poses ATP1 and ATP2 (Supplementary Fig. 14a–d). During the MD simulations molecules placed at ATP2 remained bound at their starting locations (the RMSDs of the nucleotide atoms were around 2 Å; Supplementary Fig. 14d; Movie 2). Molecules at ATP1 shifted from their initial location (Supplementary Fig. 14c; Movie 1). Taken together, the simulations performed suggest that although pose M1 is favorable for MANT-GTP, a similar pose for ATP would be unfavorable, which is consistent with the previous studies. It is likely that MANT-GTP, an AC inhibitor, is stabilized in the corresponding M1 conformation through additional π–π stacking interactions of the MANT group (e.g., with the W1188 residue in AC9).

**AC9 conformations stabilized by distinct activators**. The availability of AC9 structures captured in distinct conformations elicited by different activating agents (Gαs protein, DARPin C4, forskolin) allows us to determine the elements of AC9 structure that respond to the interaction with these activators (Figs. 4, 5). We aligned AC9–Gαs, AC9–C4–M, AC9$_{1250}$–Gαs-M, AC9$_{1250}$–Gαs–MF to AC9–M, using the C2a domain as an alignment template (Fig. 4, Movie 3). The binding of DARPin C4 to the C2a domain of AC9

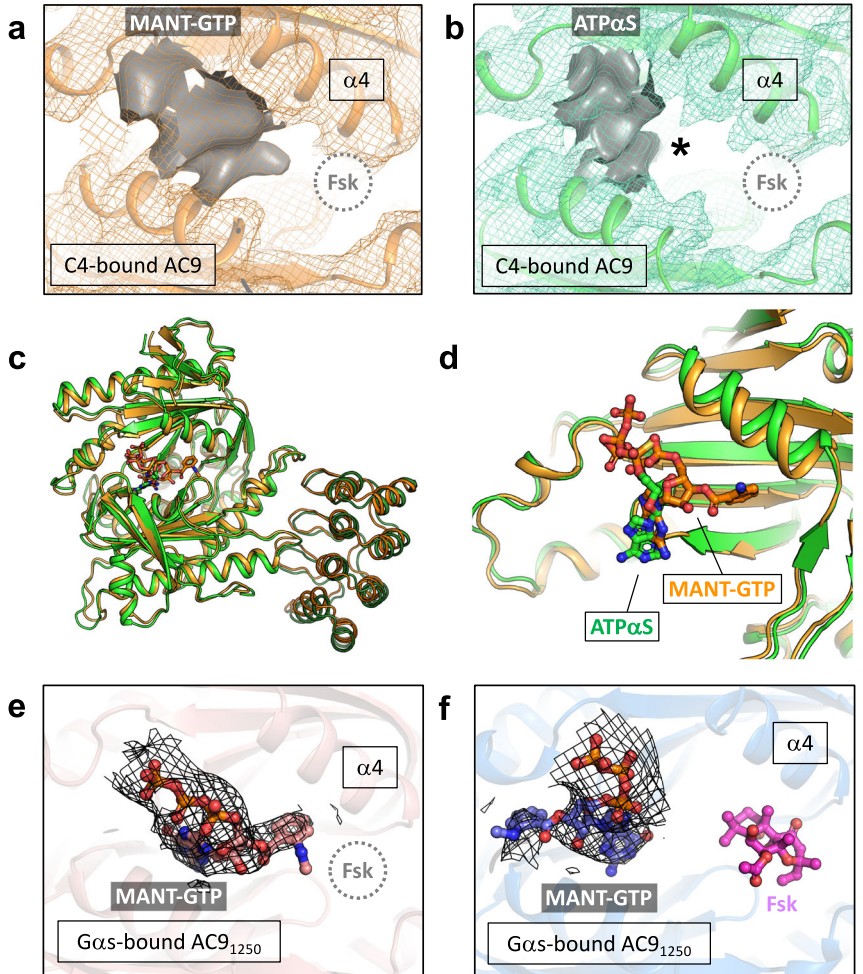

**Fig. 3 Experimentally observed density in the nucleotide-binding site of AC9. a** The portion of the density corresponding to the bound nucleotide (MANT-GTP) in the map AC9–C4 is shown as gray surface; the position of the forskolin binding site is indicated with a dashed circle. **b** A similar view of the AC9–C4 in complex with ATPαS. The asterisk indicates the absence of extended density. **c** Alignment of the soluble domains of AC9 in a complex with C4 and MANT-GTP (orange) or ATPαS (green) reveal a low RMSD of 1 Å. **d** Although the conformation of the catalytic domain is very similar, the pose of the nucleotide is different based on the density (as in (**a**, **b**)), with MANT-group of MANT-GTP pointing towards the forskolin site. **e** The density map corresponding to the bound MANT-GTP molecule in the AC9$_{1250}$–Gαs–M structure (forskolin-free). The density is similar to that in (a). **f** The density corresponding to MANT-GTP in the AC9$_{1250}$–Gαs–MF structure is shown as black mesh. The molecule of forskolin (Fsk) is shown as magenta sticks.

leads to a small rearrangement of the C1a domain (RMSD ~1.2 Å), particularly prominent at the flexible "claw" region (β6α5β7β8) of C1a domain (Fig. 4b). Close to the "claw" region, the helix α4 in the AC9–C4–M structure also moves outward (RMSD ~1.7 Å) compared to AC9. A similar conformational change is also observed in the AC9–C4–A structure, indicating that the overall domain rearrangement is primarily driven by the DARPin C4 rather than by the ligand occupying the active site of the cyclase (Supplementary Fig. 15). The helix α4 lines the forskolin binding interface and has direct interactions with forskolin. In the AC9–Gαs structures, this α4 helix is displaced further (Fig. 4c, d), with RMSD values of 3.15 Å in AC9$_{1250}$–Gαs–M and 5.7 Å in AC9$_{1250}$–Gαs–MF. Likewise, the helix α4 in the occluded state of AC9–Gαs shows an RMSD of 5.0 Å, compared to the AC9–M structure (Fig. 4a). This comparison establishes a clear pattern of conformational changes that correlate with the activation state of the AC9, where helix α4 appears to move out and create an opening in the allosteric site of AC9 concomitantly with the increase in the catalytic activity of the enzyme (Figs. 4, 5). This opening is relatively small in the partially activated state (AC9–C4), and becomes substantial in the forskolin-free, G protein-bound state of AC9 (AC9$_{1250}$–Gαs–M). It is likely to be further stabilized in the presence of forskolin (AC9$_{1250}$–Gαs–MF).

## Discussion

DARPins are established as powerful agents for stabilization of macromolecules for structural and biochemical studies[20,28]. Although our primary motivation to generate DARPins that bind AC9 was to facilitate AC9 structure determination, the discovery of DARPin C4 as a partial AC9 activator has provided us with a unique opportunity to probe the structure and function of AC9. This reagent has proven to be an excellent tool for studying the molecular determinants of AC9 activation and autoinhibition. The inability of DARPin C4 to stablize the occluded state of AC9 points to a deep link between the specific activation of AC9 by the G protein αs subunit and the built-in autoregulatory mechanism for controlling the cAMP generation by this membrane enzyme. This autoregulation may be phosphorylation-dependent, as suggested by a recent mutagenesis-based study of AC9 autoinhibition by its C-terminus[24]. It is possible that the difference in the interaction interface between DARPin C4 and Gαs accounts for the ability of the G protein to trigger occlusion of the active and allosteric site by the AC9 C-terminus. It is also possible that the disordered regions of AC9, invisible in our 3D reconstruction, play a role in the enzyme activation and our results hint at a yet undescribed interplay between the unstructured loop regions

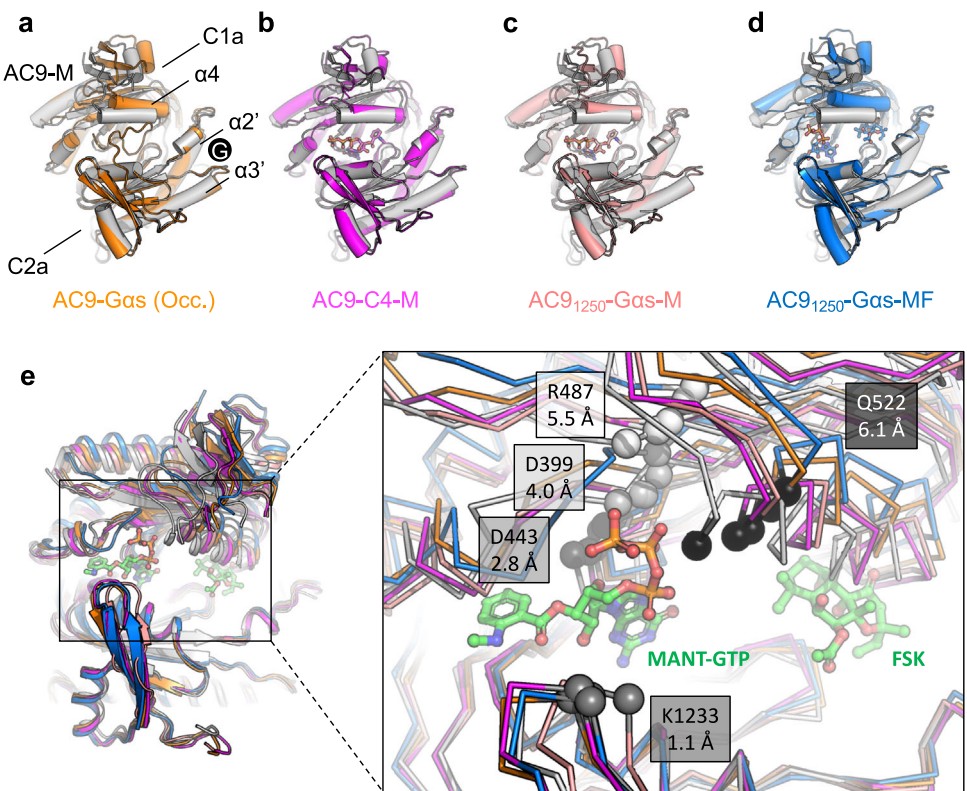

**Fig. 4 Structural transitions corresponding to the distinct activator-bound states of AC9. a** The AC9–M structure (white) was structurally aligned to AC9–Gαs (orange) using the C2a domain as an anchor. The helices α4, α2' and α3' are indicated in the panel. The position of Gαs protein binding is indicated with a black circle. **b–d** Same as (**a**), with the AC9 bound to DARPin C4 (**b**; magenta), AC9$_{1250}$–Gαs-M (**c**; pink), and AC9$_{1250}$–Gαs–MF (**d**; blue). **e** Comparison of the five available structures reveals relative displacement of the active site residues, D399, D443, R487, and K1233, as well as the Q522 residue in the helix α4 adjacent to the forskolin site. The rearrangement of the active site residues proceeds in a manner that correlates with the activation state of the protein (additionally illustrated by the morphs in Movie S3). Inset: the values of Cα atom displacement from the AC9–M (white ribbon) to AC9$_{1250}$–Gαs–MF (fully activated state, blue ribbon) are indicated in gray boxes (Å).

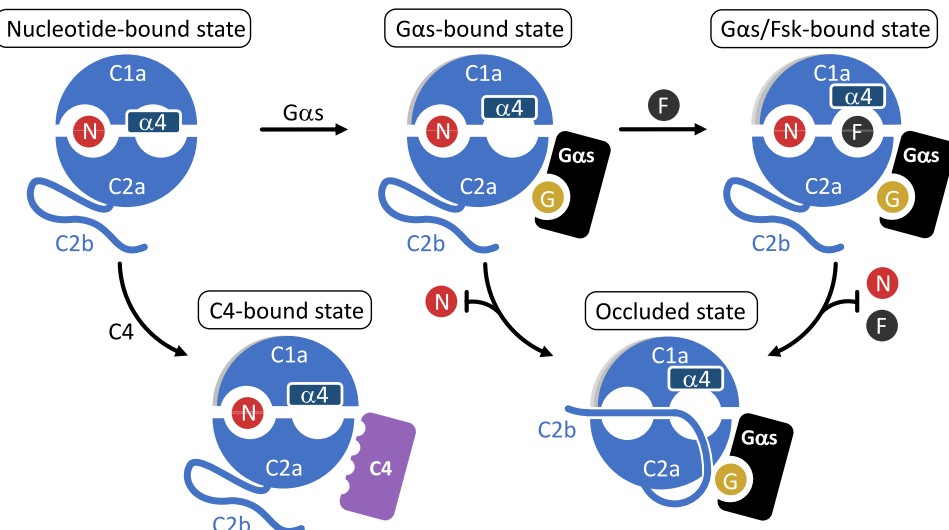

**Fig. 5 Conformations observed in the available AC9 structures.** Based on the available structures, the G protein-free state (nucleotide-bound state, corresponding to AC9–M) can transition to a partially active state upon Gαs binding (Gαs-bound state); subsequent binding of forskolin (Fsk) fully activates AC9 (Gαs/Fsk-bound state). The G protein-activated states can be inhibited by the C2b domain of AC9 with formation of the occluded state. A partially active state can be formed upon DARPin C4 binding. This conformation of AC9 corresponds to a partially activated state, but it is a state that does not favor the formation of the stable occluded conformation, based on the structural evidence. Circles labeled "N", "G" and "F" correspond to ATP or an ATP substitute (MANT-GTP or ATPαS), GTP and forskolin, respectively. The dark-blue rectangle corresponds to the helix α4, adjacent to the forskolin binding site. The stop-arrows indicate inability of the nucleotide and/or forskolin to bind to the AC in the occluded state.

(e.g., N- and C-terminus, C1b region) of the membrane ACs and their regulators, such as the Gαs protein.

The prominent conformational changes involving the helix α4 of AC9 draw parallels to the mechanism of soluble AC (sAC) activation[29]. In sAC, bicarbonate binding to the allosteric site (approximately corresponding to the forskolin site in the membrane ACs) induces the conformational change of the side-chain of R176 residue located on α4. The rearrangement of R176 releases the side-chain of D99, acting as a switch that enables the formation of the catalytic cation-binding site (Supplementary Fig. 16). The helix α4 in sAC and its equivalent in AC9 appears to be linked the enzymatic activation mechanism. Activation of sAC is controlled by minute side-chain rearrangement in R176 in response to its activator (bicarbonate). In contrast, AC9 reacts to its activators with whole domain rearrangements, with the displacement of helix α4 corresponding to the potency of the activator. The relatively low resolution of our cryo-EM structures could not allow us to determine the side-chain orientations in α4 unambiguously, and the precise residues in control of the activation mechanism will require detailed future analysis at higher resolution.

The structures of AC9 available to date have several recognized limitations: (i) The majority of the structures were obtained at a relatively low resolution (~4 Å); (ii) Many of the structures could only be determined in the presence of MANT-GTP, a nucleotide-derived AC inhibitor. The similarity of the MANT-GTP and ATPαS-bound state of AC9–C4 complex indicates that comparisons of distinct MANT-GTP-bound states provide a valid interpretation of the conformational transitions stabilized by different AC9 activators in the presence of MANT-GTP as a nucleotide substitute, at least at the resolution of our 3D reconstructions. Currently, we are severely limited in our ability to probe the structure and function of membrane ACs due to their modular organization and a high degree of flexibility. Thus, the use of minimally modified ATP analogs that closely match the structure of ATP as well as the development of novel strategies for AC9 stabilization for structural studies will be required to provide insights into the mechanism of AC9 activation at atomic resolution.

Our experiments establish DAPRin C4 as a highly valuable molecular tool that can be used for in vitro biochemistry or in cell-based assays. This reagent can not only be used to obtain unique insights into AC9 activation, but it may provide opportunities for probing distinct aspects of the cAMP signaling pathway by directly modulating the AC9 activity. Although we limited our experiments to the proof-of-concept assays in HEK293 cells, it is possible that DARPin C4 can be used in a wide variety of cellular contexts, in diverse tissues and at the whole-organism level. The properties of DARPin C4 may need to be modified for optimal performance in the specific application. For example, targeting DARPin C4 to the membrane by a lipid- or peptide-based anchor may dramatically increase its potency in vivo. Different methods of DARPin C4 delivery to the cytosol should also be explored. Nevertheless, as DARPins are highly amenable to protein engineering, DARPin C4 may be a good start for generating similar reagents with new AC selectivity profiles, which may find use in a wide range of applications.

## Methods
**DARPin generation**. To generate DARPin binders against AC9 and the biotinylated C2a domain alternating on either MyOne T1 streptavidin-coated beads (Pierce) or Sera-Mag neutravidin-coated beads (GE) depending on the particular selection round. Ribosome display selections were performed essentially as described[30], using a semi-automatic KingFisher Flex MTP96 well instrument. The DARPin library included a mix of N3C-DARPins with randomized and non-randomized N- and C-caps[20,22], respectively, and the successively enriched pools were cloned as intermediates into a ribosome display specific vector[31]. Selections

were performed over four rounds with decreasing target concentration and increasing washing steps to enrich for binders with high affinities. The final enriched pool was cloned as fusion into a bacterial pQE30 derivative vector with a N-terminal MRGS(H)₆ and a C-terminal FLAG tag via unique BamHI x HindIII sites containing lacIq for expression control. After transformation into E. coli XL1-blue, 380 single DARPin clones were expressed in 96-well format and lysed by the addition of B-PER Direct detergent supplemented with 0.4 mg/ml lysozyme and 20 U/ml nuclease (Pierce). These bacterial crude extracts of single DARPin clones were subsequently used in a Homogeneous Time Resolved Fluorescence (HTRF)-based screen to identify potential binders. Binding of the FLAG-tagged DARPins to the streptavidin-immobilized biotinylated C2a domain of AC9 was measured using FRET (donor: Streptavidin-Tb cryptate, 6.6 nM (610SATLB, Cisbio), acceptor: MAb Anti FLAG M2-d2, 6.6 nM (61FG2DLB, Cisbio)). FRET signals were recorded using a EnVision Multimode Plate Reader (Perkin Elmer). From the identified binders, 32 clones were sequenced and 21 single clones identified. To investigate binding to the full-length AC9 an additional ELISA with immobilized AC9 and AC9–Gαs was performed using bacterial crude extracts.

The DARPins were expressed in small scale, lyzed with Cell-Lytic B (Sigma) and purified using a 96-well IMAC plate (HisPur™ Cobalt plates, Thermo Scientific). After IMAC purification, DARPins were analyzed at a concentration of 10 μM on a Superdex 75 5/150 GL column (GE Healthcare) using an Akta Micro system (GE Healthcare) with PBS containing 400 mM NaCl as the running buffer.

**Protein expression and purification**. The methods for bovine adenylyl cyclase 9 (AC9, Uniprot ID E1BM79_BOVINE, with a C-terminal YFP-twinStrep tag) expression and purification were similar to those previously described[1]. Briefly, AC9 was expressed using the BacMam system[32]. HEK293F cells were grown in suspension at 37 °C to a density of (~2–2.5) × 10⁶ ml⁻¹ in Protein Expression Medium (PEM). The cells were collected by centrifugation and resuspended in Freestyle 293 expression medium. The P2 baculovirus for AC9 expression was added to the cells, followed by an incubation for 3–5 h. The PEM medium was added back to the cell suspension and the cells were further grown for ~60 h. For AC9 purification, the cell pellets were resuspended in buffer A (50 mM Tris-HCl, pH 8.0, 150 mM NaCl) supplemented with protease inhibitors (1 mM benzamidine, 1 μg/ml leupeptin, 1 μg/ml aprotinin, 1 μg/ml pepstatin, 1 μg/ml trypsin inhibitor and 1 mM PMSF). Cells were lysed using a Dounce homogenizer and the total cell membranes were collected by ultracentrifugation with a rotor Ti45 at 35,000 rpm (142,000 × g), for 1 h. The membranes were solubilized by adding 1% DDM and 0.02% cholesteryl hemisuccinate (CHS). After the second round of ultracentrifugation, the supernatant was mixed with 1 ml of CNBr-activated Sepharose coupled with anti-GFP nanobody[33], followed by a 30 min incubation with rotation. The resin was collected and washed with 40 column volumes of buffer B (50 mM Tris-HCl, pH 8.0, 150 mM NaCl, 0.1 % digitonin). The protein was eluted by adding HRV 3 C protease (1:10 w/w) overnight. The eluted protein was concentrated using a 100 kDa cutoff Amicon concentrator and applied to the Superose 6 Increase 10/300 GL column pre-equilibrated with buffer B. The peak fractions corresponding monomeric AC9 were concentrated and used for cryo-EM grid preparation.

The C-terminally truncated bovine AC9₁₂₅₀ (residues 1-1250 of AC9) was expressed using transient transfection. HEK 293 F cells were grown in suspension at 37 °C to a density of (~2–2.5) × 10⁶ ml⁻¹ in Protein Expression Medium (PEM). The cells were collected by centrifugation and resuspended in Freestyle 293 expression medium. The transfection mixture was prepared containing plasmid and polyethyleneimine (linear PEI max, Polysciences) at a ratio of 1:3 (w/w). After a 3–5 h incubation, PEM medium was added back to the cells and the cells were cultured for ~60 h. The purification procedure for AC9₁₂₅₀ was identical to that for AC9.

The procedure for expression and purification of the bovine Gαs (Uniprot ID P04896-1, with a C-terminal 8xHis-tag) was similar to the one previously described[1]. The standard Bac-to-Bac baculovirus expression system was used (Invitrogen): High Five insect cells were cultured in suspension (1 L) to a cell density of 1.5 × 10⁶ ml⁻¹ and the P2 virus was added to infect the cells for protein expression. The cells were harvested after 72 h. For protein purification, the cells were resuspended in buffer A, lysed using a Dounce homogenizer and solubilized by adding 1% dodecyl-β-maltoside (DDM) for 1 h. The lysate was clarified by ultracentrifugation. The supernatant was incubated with 1 ml Ni-NTA resin for 30 min. The resin was washed with buffer C (50 mM Tris-HCl, pH 8.0, 150 mM NaCl, 0.02 % DDM) supplemented with 20 mM imidazole. This was followed by a second wash with buffer C supplemented with 40 mM imidazole. The protein was eluted with buffer C supplemented with 250 mM imidazole. The eluted protein was concentrated using a 10 kDa Amicon concentrator and applied to a Superdex 200 Increase 10/300 GL column pre-equilibrated with buffer B. The peak fractions were concentrated, snap-frozen in liquid nitrogen in aliquots, and stored at −80 °C until the day of the experiment.

BL21(DE3) cells were transformed with a plasmid encoding a DARPin of interest with an N-terminal 8xHis-tag and a C-terminal Flag-tag. The cells grown overnight, and 1 L of LB medium was inoculated and grown at 37 °C to an OD₆₀₀ of 0.8. To induce protein expression, 1 mM IPTG was added to the culture and the induced cells were further grown at 37 °C for 3 h. After harvesting by centrifugation, the cells were resuspended in buffer A supplemented with 1 mM PMSF and lysed by

sonication. The lysate was clarified by centrifugation and the supernatant was incubated with 1 ml Ni-NTA resin for 30 min. The resin was washed with buffer A supplemented with 20 mM imidazole, followed by a second wash with buffer A supplemented with 40 mM imidazole. The protein was eluted using buffer A supplemented with 250 mM imidazole, applied to a Superdex 200 Increase 10/300 GL column pre-equilibrated with buffer A. The peak fractions were concentrated, snap-frozen in liquid nitrogen in aliquots, and stored at −80 °C until the day of the experiment.

For purification of the AC9–C4 complex, upon elution of AC9 from the Sepharose-GFP nanobody resin, the protein was mixed with the purified DARPin C4 at a molar ratio of 1:2. The complex was incubated for 30 min at 4 °C and applied to the Superose 6 Increase 10/300 GL column pre-equilibrated with buffer B. The fractions corresponding to the AC9–C4 complex were collected and concentrated for cryo-EM grid preparation.

**Cryo-EM sample preparation.** For AC9–M, AC9–C4–M and AC9–C4–A, the solutions containing the purified AC9 or AC9–C4 at a concentration of about 5 mg/ml were incubated with 5 mM MnCl₂, 0.5 mM MANT-GTP (0.5 mM ATPαS for AC9–C4–A sample) for 30 min at 4 °C. A small aliquot of the sample (3.5 μl) was applied to the glow-discharged UltraAuFoil 1.2/1.3 300-mesh grid. The grid was blotted for 3 s, plunge-frozen in the liquid ethane using Vitrobot Mark IV (Thermo Fisher Scientific). The grids were transferred to and stored in liquid nitrogen for subsequent cryo-EM data collection.

For AC9₁₂₅₀–Gαs–M, the purified AC9₁₂₅₀ at a concentration of 5 mg/ml was incubated with 5 mM MnCl₂, 0.5 mM MANT-GTP and a 2-fold molar excess of GTPγS-activated Gαs protein for 30 min at 4 °C. A small aliquot of the sample (3.5 μl) was applied to the glow-discharged Quantifoil R1.2/1.3 200-mesh grid. The grid was blotted for 3 s, plunge-frozen in the liquid ethane using Vitrobot Mark IV (Thermo Fisher Scientific). The grids were transferred to and stored in liquid nitrogen for subsequent cryo-EM data collection.

**Cryo-EM data collection and processing.** The cryo-EM datasets were collected using a Titan Krios electron microscope equipped with a K2 Summit direct electron detector (Gatan) and a GIF-Quantum energy filter (slit width of 20 eV) at EMBL Heidelberg. The micrographs were recorded using SerialEM[34] in super-resolution mode. The defocus range was set from −0.75 μm to −2.5 μm. Each micrograph was dose-fractionated to 40 frames with a total exposure time of 8 s, resulting in a total dose of about 40 e⁻/Å².

The movies were binned two-fold and motion-corrected using MotionCor2[35], yielding micrographs with a pixel size of 0.81 Å. Gctf[36] was used for CTF estimation. About 2000 particles were first manually picked and 2D classified in Relion 3.0[37]. The selected 2D classes were used as templates to autopick the particles in all selected micrographs (AC9: 8608 micrographs, AC9–C4: 8481 micrographs, AC9₁₂₅₀–Gαs–M: 10312 micrographs). The initial model of AC9 was generated from PDB: 2HYD [https://www.rcsb.org/structure/2HYD]using EMAN2[38] and lowpass filter to 40 Å. The refined map of AC9 was used as the initial model for 3D reconstruction of AC9–C4. For the 3D reconstruction of AC9₁₂₅₀–Gαs–M, the map of AC9–Gαs (EMD-4719) was used as the initial model. In each case, after several rounds of 2D and 3D classifications the best 3D classes were selected and 3D refinement and post-processing were performed in Relion3.0. Local resolution maps were calculated by ResMap[39] implemented in Relion 3.0. The detailed steps of image processing for each dataset are shown in Supplementary Fig. 1, Supplementary Figs. 6, 7, 9 and Supplementary Table 1.

The AC9–C4–A dataset was collected using Titan Krios electron microscope equipped with a K3 direct electron detector at ScopeM at ETH Zurich. The micrographs were recorded using EPU in super-resolution mode. The defocus range was from −1 μm to −2 μm. Each micrograph was dose-fractionated to 40 frames with a total exposure time of 0.7 s, resulting a total dose of about 48 e⁻/Å². All data processing was performed according to the procedure described above for AC9–C4–M.

**Model building and refinement.** Model building was carried out manually in COOT[40]. The model of AC9–Gαs (PDBID: 6R3Q [https://www.rcsb.org/structure/6r3q]) was used as reference. The AC9 domains M1, M2, C1a, and C2a were fitted individually according to the cryo-EM map of each dataset. The side chains of the amino acids were adjusted based on the clearly defined densities of the bulky residues (Phe, Trp, Tyr, and Arg). MANT-GTP was built using COOT. The model was refined using real_space_refine in PHENIX[41]. Each model was validated as previously described[1]. Briefly, the atoms of the final models were randomly displaced by 0.5 Å using PDB tools implemented in PHENIX. The perturbed models were refined in PHENIX against the half map1. The refined models were used to generate FSC of model versus half map2. The map vs model FSC curves agreed well for all structures. The geometries of the models were validated using MolProbity[42]. All figures were prepared in PyMol[43] or Chimera[44].

**Molecular dynamics simulation.** The catalytic domains of AC9, AC9₁₂₅₀, and AC9₂₂₅₀, in complex with Gαs (Gαs–C1a–C2a) were used for molecular dynamics (MD) simulations. The simulations were performed using the AC9₁₂₅₀–Gαs–M model (in the absence of forskolin). Two ATP conformations (A1 and A2) were

manually built-in COOT, based on the density elements corresponding to MANT-GTP in the structure AC9₁₂₅₀–Gαs–MF (PDB: 6R4O [https://www.rcsb.org/structure/6R4O]) and in the crystal structure AC5₍C1₎–AC2₍C2₎–Gαs–ATPαS (PDB:1CJK [https://www.rcsb.org/structure/1CJK]) and AC5₍C1₎–AC2₍C2₎–Gαs–ATP (PDB: 3C16 [https://www.rcsb.org/structure/3C16]). The MD system was prepared using a solution builder in CHARMM-GUI[45]. The model was neutralized by adding ions (with the Monte-Carlo method). The water box type was rectangular with an edge distance of 20 Å. The system was solvated with TIP3P water molecules. The CHARMM36m force field was used for the simulations and Gromacs 2019.2[46] was used as the MD engine. The energy minimization was performed with a Fmax tolerance of 1000 kJ mol⁻¹ nm⁻¹. The system was equilibrated for 125 ps. The trajectory was accumulated for 50 ns with a timestep of 2 fs. The trajectory data were analyzed by VMD[47] and visualized using VMD and PyMOL.

**Adenylyl cyclase activity assay.** Assay of purified adenylyl cyclase activity was performed as previously described[1,48]. The reaction system contained 50 mM Tris-HCl, pH 7.5, 150 mM NaCl, 0.1% digitonin, 5 mM MnCl₂, 100 μM total ATP (10 nM [³H] ATP, PerkinElmer NET1189001MC) and 0.01 mg/ml purified AC9. For DARPin characterization, the purified DARPin protein was added to the system to a final concentration of 0.66 μM (corresponding to an AC9 to DARPin molar ratio of 1:10). The reaction was initiated by adding ATP. The reactions were performed at 30 °C for 10–30 mins and terminated by the addition of 20 μl 2.2 M HCl and incubation at 95 °C for 4 min. The samples were transferred to disposable columns filled with 1.3 g aluminum oxide. [³H] cAMP was eluted with 4 ml ammonium acetate (0.1 M) and determined using liquid scintillation counting.

The effect of DARPin C4 on human AC5, human AC6, bovine AC8 and *M. tuberculosis* Rv1625c, in comparison to bovine AC9 and AC9₁₂₅₀, was assessed by preparing two reaction mixtures per tested AC: in the presence and in the absence of 10 μM DARPin C4. The final concentration of 0.005 mg/ml was used for AC5, AC6 and AC8; 0.01 mg/ml concentration was used for AC9 and AC9₁₂₅₀, and 0.001 mg/ml concentration was used for Rv1625c.

For dose–response curves for DARPin C4- and Gαs-mediated activation of AC9, DARPin C4 or Gαs proteins were used in a final volume of 200 μl at the following concentrations: 0 μM, 0.033 μM, 0.1 μM, 0.33 μM, 0.5 μM, 1 μM, 3.3 μM, and 10 μM.

For forskolin dose–response curves, AC9 (0.005 mg/ml) and AC9 + DARPin C4 (at a ratio of 1:2) were used in the reaction. Forskolin was added at the following concentrations: 0 μM, 1 μM, 3.3 μM,10 μM, 33 μM, 100 μM, 330 μM, 1 mM.

Membrane assay of AC activity from Sf9 cells expressing individual AC isoforms was performed as described[49]. AC isoforms were stimulated with 50 nM DARPin and/or 50 nM Gαs in the presence of 5 mM Mg²⁺ and 200 μM ATP. DARPin dose–response curves for AC9 were performed in the presence of 5 mM Mg²⁺ or Mg²⁺ plus Mn²⁺ (0.5 mM) and the indicated concentrations of DARPin.

All data were analyzed using GraphPad Prism[50]. All enzymatic activity assays were performed at least three times (n = 3) unless indicated otherwise. The values were compared using one-way analysis of variance (ANOVA), followed by Tukey's multiple comparisons test implemented in GraphPad Prism.

**Cell culture for in vivo cAMP accumulation assays and FRET analysis.** To study the interactions of DARPin C4 with AC9 by Förster Resonance Energy Transfer (FRET) microscopy, to determine the effects of this reagent on the in vivo AC activity of AC9, as well as test expression levels of AC9 compared to AC9 co-expressing DRAPin C4 by in gel fluorescence imaging and fluorescence SEC, 1 × 10⁶ cells were seeded per well in a 6-well plate in DMEM medium supplemented with 5% FCS and penicillin/streptomycin. To analyze expression levels of AC9 in the presence or absence of DRAPin C4 by confocal fluorescence microscopy, 30 000 cells were seeded in poly-L-lysine pre-coated ibidiTreat μ-slide 8-well chambers in DMEM supplemented with 5% FCS penicillin/streptomycin.

The cells were cultured overnight at 37 °C and 5% CO₂ and the next day co-transfected with the plasmids of interest using PEI, at a DNA to PEI ratio of 1:2 (w/w), with the equal amounts of the AC9–YFP and DARPin-C4-CFP plasmids. In the case of in vivo cAMP accumulation assay and all expression level tests, the negative controls included cells transfected with pcDNA3.1 alone, as well as with the individual plasmids encoding AC9–YFP, DARPin-C4-CFP alone. For FRET analysis, the serotonin transporter SERT, tagged with an N-terminal CFP and a C-terminal YFP, C-SERT-Y[51], was used as the membrane FRET control, CFP–YFP fusion as cytoplasmic FRET control and YFP-SERT, YFP-SERT with DARPin C4–CFP, DARPin C4–CFP and AC9–YFP were used as negative controls. Whenever cells were transfected with only one test plasmid, pcDNA3.1 was used in 1:1 ratio [w/w] to maintain uniform DNA amounts across all samples. For the transfection procedure, DNA and PEI dilutions were prepared separately, mixed and incubated for 5–10 min before they were added to the cells. In the case of the in vivo cAMP accumulation assay, the medium was additionally supplemented with 500 μM isobutyl-methylxanthine (IBMX) and the cells were incubated at 37 °C and 5% CO₂ for 48 h.

For AC9 expression level tests, HEK293 cells were cultured in 6-well plates with 700,000 cells per well overnight. The cells were transfected with AC9–YFP, DARPin C4–CFP, AC9–YFP + DARPin C4–CFP, and pcDNA3.1 plasmids. After two days, the cells were collected and washed with PBS. The cells were resuspended in buffer A and lysed by sonication, followed by solubilization with 1% DDM-CHS at 4 °C for 1 h. The sample was clarified by centrifugation (Eppendorf 25,000 rcf/

min). For in-gel fluorescence analysis, 10 μl aliquots of the supernatant from each sample were loaded onto a 4–10% gradient SDS-PAGE (Bio-Rad) and imaged using AI600 imaging system (GE). For fluorescence size exclusion chromatography, 50 μl of each sample were analyzed using HPLC with an Agilent Bio SEC-5 500 Å column. The peak was monitored at 527 nm. The data were exported and analyzed using GraphPad Prism.

**In vivo cAMP accumulation assay**. The medium was aspirated from each well, and the cells were washed with 1 ml of PBS. Cells were lysed by the addition of 1.5 ml of lysis and detection buffer (Cisbio) for 30 min. Subsequently, 16 μl aliquots of the lysate were added to the wells of a 384-well plate (PerkinElmer) in triplicates, together with 4 μl of pre-mixed HTRF antibodies (cAMP Gs dynamic kit, Cisbio) and the plate was sealed and incubated at room temperature for 1 h. The HTRF was measured at 620 nm and 665 nm using PheraSTAR® FSX plate reader (BMG Labtech) using the TR-FRET optical module. The HTRF ratio and cAMP concentration were calculated according to the cAMP Gs dynamic kit protocol.

**FRET and confocal microscopy**. The day before FRET experiments, 120,000 cells were seeded into ibidiTreat μ-slide 8-well microscopy chamber (ibidi) and allowed to adhere overnight. Before microscopy experiments, the medium was replaced with FluoroBrite™ DMEM medium supplemented with 5% FCS and penicillin/streptomycin.

For FRET experiments, the cells were imaged using Nikon Ti Eclipse epifluorescence microscope equipped with an ORCA Flash 4.0 camera (Hamamatsu) using a Nikon CFP Plan Apochromat 60x, NA 1.4 objective at 37 °C and 5% $CO_2$. The "three-filter method" of FRET analysis was performed using a SpectraX light engine, equipped with CFP, YFP and FRET excitation and emission band-pass filters (CFP excitation at 430/24 nm, emission at 470/24 nm; YFP excitation at 500/20 nm, emission at 535/30 nm; FRET excitation at 430/24 nm, emission at 535/30 nm) with 200 ms exposure. The images were processed using Image J plugin PixFRET (version 1.8.0_202). The background (BG) and spectral bleed-through (SBT) parameters were determined for the donor and acceptor separately and were used to generate the calculated FRET and FRET efficiency images. The calculated BG values were subtracted from the final donor and acceptor images. The greyscale FRET efficiency image was used to measure FRET efficiency values at the plasma membrane, which was predefined as the region of interest, except for CFP–YFP construct, which is expressed as a cytosolic protein. A line was drawn randomly through the plasma membrane and pixel intensities along the line were plotted. The pixel intensity values from the plasma membrane were averaged and the average used as final FRET efficiency value per cell. FRET efficiencies were expressed as mean ± S.E.M.

For microscopy expression level tests, the cells were imaged using Leica Stellaris confocal microscope (LAS X 4.3.0.24308) equipped with a HyD detector in a frame sequential data acquisition mode, using Diode 405 and OPSL488 excitation lasers, and a Leica HC PL APO 63x immersion oil objective (NA 1.4) at room temperature. Seven images were collected per condition for each experiment at 16-bit depth and 2048 × 2048 pixel sampling rate. The images were corrected for SBT and BG using LUMoS spectral unmixing plug-in in Fiji (ImageJ)[52]. The YFP channel image was then converted to 32-bit depth and the threshold was defined to determine the pixels which correspond to cellular membranes using the default method of threshold determination and auto-adjustment. After the application of the threshold to the image, the pixel intensities were measured. The pixel intensities are represented as mean ± S.D. The results were compared using one-way ANOVA followed by Dunnett's multiple comparisons test. All statistical analysis was performed using GraphPad Prism 8.0.0.

**Reporting summary**. Further information on research design is available in the Nature Research Reporting Summary linked to this article.

## Data availability statement
The cryo-EM density maps have been deposited in the Electron Microscopy Data Bank, with accession numbers EMD-13330, EMD-13331, EMD-13334, EMD-13335, EMD-13336, EMD-13337 and EMD-13338. The coordinates have been deposited in the Protein Data Bank, with entry codes PDB:7PD4, PDB:7PD8, PDB:7PDD, PDB:7PDE, PDB:7PDF, PDB:7PDG and 7PDH. All materials are available from the corresponding author upon reasonable request. Source data are provided with this paper.

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

## Acknowledgements

The authors thank the PSI EM Facility (Emiliya Poghossian and Elisabeth Müller-Gubler) and the Electron Microscopy Core Facility at EMBL Heidelberg. We thank Felix Weis (EMBL Heidelberg) for the support in high-resolution cryo-EM data collection. We thank Spencer Bliven and Marc Caubet Serrabou (PSI) for high performance computing support. We thank Thomas Reinberg, Sven Furler, Cristian Thom and Joana Marinho for performing the ribosome display selection and screening at the HT-BSF unit at the University of Zurich. This study was supported by iNEXT, the Swiss National Science Foundation (VMK, SNF Professorship, 150665 and 176992), Horten Foundation grant (VMK), and a National Institutes of Health Grant RO1 GM060419 (CWD).

## Author contributions

C.Q. designed the experiments, performed molecular cloning, purified the proteins, collected cryo-EM data and determined the cryo-EM structures. P.L. performed the FRET experiments. V.M. and B.K. purified AC5 and AC8. I.M. contributed to cryo-EM data collection. C.Q., P.L., Y.L performed adenylyl cyclase activity assays. C.Q., M.L, and A.N.B performed MD simulation. J.V.S, B.D., and A.P. generated the DARPins. C.W.D. designed and supervised activity assays. V.M.K. designed the experiments, initiated, and supervised the project. C.Q and V.M.K collected the data from all authors and prepared the manuscript. C.Q., P.L., A.N.B., A.P., C.W.D., and V.M.K. contributed to writing the manuscript.

## Competing interests

The authors declare no competing interests.
