## [Peer Review File · Nature Communications]

Structural basis of adenylyl cyclase 9 activationEditorial Note: This manuscript has been previously reviewed at another journal that is not operating a transparent peer review scheme. This document only contains reviewer comments and rebuttal letters for versions considered at *Nature Communications*.

REVIEWER COMMENTS

Reviewer #1 (Remarks to the Author):

NCOMMS-21-33679-T

Structural basis of adenylyl cyclase 9 activation

This revised manuscript is improved from the original [redacted] submission; however, there remain a few substantive concerns. In the introduction, the authors cite controversy surrounding AC9 activation by forskolin, without including the (at least partially) enlightening study demonstrating that that mouse AC9 lacked key residues essential for forskolin activation (Yan et al Mol Pharm 1998).

As cited in my initial review, the rationale for including the molecular dynamic simulations was unclear. The authors wisely pursued and succeeded in obtaining additional structures which clarify the issue of ATP binding, which renders the molecular dynamic simulations even more extraneous in this revised manuscript. Without better justification, it remains unclear why they are included in this manuscript.

Finally, as suggested by the reviews of the previous submission, the authors included discussion of the other mammalian adenylyl cyclase whose structures are known in both active and inactive states (soluble adenylyl cyclase; ADCY10). However, it is unclear why they exclusively focus on the structure of the ATP bound in the sAC structure bound to its inhibitor LRE1. There are structures of sAC bound to ATP analogs which do not include inhibitor, which would be a more relevant structure to use for comparison to an “active” AC9 structure.

Reviewer #2 (Remarks to the Author):

In the revised manuscript, entitled 'Structural Basis of adenylyl cyclase 9 activation' by Qi et al, the authors made several modifications that improved the manuscript. In many ways the real novelty of this study is the novel partial agonist, C4. However, in this revised manuscript the authors maintained the description and discussion of the alternative conformational states of AC9 stabilized by MANT-GTP, despite reviewers' sharp criticism of the relevance of MANT-GTP as a model for ATP binding. The AC field has evolved significantly through the last decade or so and the biochemical and structural data describing MANT-GTP as a unique inhibitor are plentiful. There is a tendency for the authors to obfuscate the potential binding sites for ATP based on binding poses of the inhibitor MANT-GTP. As mentioned previously, the MANT moiety itself presents confounding effects on AC structure that ATP would never adopt. The fact that the triphosphates of MANT-GTP are not even poorly hydrolyzed would support this. In addition, the capacity of the MANT moiety to interfere with the presumptive Mg²⁺ coordination, and therefore the contributions of the highly conserved and highly important two aspartate residues in the C1 domain, argue that the fluorescent GTP molecule is not a good model for ATP. In all, the authors might be better off diminishing the emphasis on MANT-GTP as a model for alternative conformation binding states of ATP and enhance the description of the structure and mechanism of the novel partial agonist C4.

Comments:

The real novelty of this study is the identification and structure of the C4 DARPin partial agonist. The authors include extra data on catalytic constants. Note that the authors should include the V_{max} values of AC9 in the presence of Galphas and Mn in their description of progress curves in the presence of C4. While only 4-fold higher than basal activity, the V_{max} in the presence of Gs is approximately 24-fold higher than basal and 7.5 higher than C4 (assuming V_{max} with Gs ~ 900 nm/min/mg). The magnitude of the partially agonist activity needs to be put into perspective.

The 'in cellulo' measurements (not in vivo) of C4 studies need to include data on whether AC9 expression was affected by C4 overexpression. Such studies would include measurements of plasma membrane expression and/or maximal adenylyl cyclase activity (Galphas with Mn²⁺). These studies could be hormone-stimulated.

The description of the mechanism of C4 activation compared to Galphas remains unsatisfying. Although both proteins engage the alpha2 and alpha3 helices of the C2 domain, only Galphas engages the C1 domain. In fact the entire N-terminal region of the C1 domain is disordered in the presence of the DARPin, highlighting its inability to full active AC9. It is understandable that the authors are trying to argue that activation of cyclase involves more than bringing the two domains together, which previous models based on the actions of forskolin have implied. However, the actions of Galphas, as a full agonist, engages both the C1 and C2 domains in a slightly different manner than forskolin, while C4, a partial

agonist engages only the C2 domain. This is quite clear from the AC9-Galphas structures and those of the C1(AC5)-C2(AC2) cytosolic domains.

The use of MANT-GTP still remains a little bit of an issue. Following the recommendations of the reviewers the authors removed state 3 from their models in substrate-bound mechanistic models. MANT-GTP is still an inhibitor and stabilizes a different conformational state than a substrate. The MANT moiety is not silent and plays a much larger role in AC9 binding than would ATP, or even cAMP, for that matter. This is extremely difficult to understand the relevance of this state to ATP sub-states, particularly when the geometry of the 3'OH on the ribose ring is quite different with MANT-GTP compared to other non-hydrolyzable ATP analogues.

The elongated density observed with the MAN-GTP extending into the forskolin binding site is interesting. This overlap would imply that forskolin binding may impair MANT-GTP binding. If the authors want to imply that the binding of MANT-GTP (and thus potentially GTP) with forskolin, then they should actually test this. This is easily testable since MANT-GTP is fluorescent. In this case fluorescence anisotropy measurements following MANT could be made in the presence and absence of forskolin (dose response curve of forskolin).

The authors appropriately raise the controversy regarding forskolin binding to AC9. As noted there do not appear to be any glaring side-chain incompatibilities in AC9 that would prevent forskolin binding, agreeing with literature reporting forskolin-mediated activation of AC9. The fact that the authors are able to delineate the structure of forskolin-bound AC9 validates this. The controversy, as implied by the authors, is likely based on how cyclase activity was measured. Most membrane-bound mammalian cyclases (but not all) display positively cooperative actions between Galphas and forskolin. The proximity of the two binding sites for these modulators makes structural sense for how this allostery could occur. Do the authors have data to measure this cooperativity exists to explain why forskolin, on its own, very poorly activates AC9 ? Forskolin dose-response measurements in the absence and presence of GalphaS would be appropriate. Such data would easily address the apparent controversy regarding forskolin-insensitivity of this isoform. Similar experiments should also be made using the partial agonist C4.

The mystery still remains why Galphas binding appears to stabilize the auto-inhibitory state in the cryoEM structure. The authors cite work on phosphorylation as a possible contributor however they need to describe, at least to speculate how phosphorylation may affect the structure of the C-terminal region of C2. This could be depiction of the putative phosphorylation sites relative to the C-term in the auto-inhibited state.

Minor comments:

In line 202 the authors refer to ATP analogue ApCpp. Are they referring to AMP-Cpp ? They should refer this.

Reviewer #4 (Remarks to the Author):

The manuscript entitled "Structural basis of adenylyl cyclase 9 activation" reports several full-length structures of the membrane protein AC9, acquired by cryo-EM, in in complex with natural or a novel synthetic activator proteins and nucleotide analogues.

The structural data are accompanied by appropriate biochemical characterization of the synthetic activator, in vitro and in vivo enzymatic assays and in vivo microscopy, in order to progress in the understanding of AC activation.

This is an impressive amount of work and it is highly appreciated having this type of study for full-length membrane proteins.

I have evaluated only the part related to MD simulation.

In this part, the authors simulated the soluble domains of AC9 in complex with the natural protein activator GalphaS and ATP bound in two different conformations:

- Conformation A1, derived from the MANT-GTP conformation M1 seen in the AC9_1250/GalphaS/MANT-GTP/Forskolin structure (previously published),
- Conformation A2, derived from the MANT-GTP in conformation M2 seen in the AC9/GalphaS/MANT-GTP structure presented here.

They observed that only conformation A2 is stable, supporting the M2 conformation of MANT-GTP.

My main recommendation would be to better articulate this analysis with the structural data presented for AC9+DARPIN-C4+ATPalphaS.

Since there are structural data with ATP bound, it is not completely clear what MD simulations bring to the study.

1/ Regarding the motivations, the authors sought to: 1) 'To determine whether AC9 may have any preference for a specific nucleotide pose' and 2) 'to ascertain that the M2 state may represent a stable nucleotide-bound conformation'.

I would think that Figure 3 already addresses both points: MANT-GTP can adopt two different conformations, depending on the presence of forskolin (or other ligand in this site), as shown in Figure 3e and 3f. In the absence of forskolin, MANT-GTP occupies a part of the forskolin binding site in the M2 conformation, stably enough to observe density at this site.

Although the stability of A2 confirms that the proposed model of MANT-GTP M2 conformation is reasonable, the MD analysis indeed raises further questions.

2/ Regarding the ATP conformations used in the simulations:

What is the agreement between the M1 pose and the density of MANT-GTP seen in AC9_1250+GalphaS+MANT-GTP ? Looking at Figures 3e and 3f/13a, it seems rather limited, which makes it a questionable starting point for a simulation with ATP in the A1 pose.

Is the A2 conformation similar to ATPalphaS in the AC9/DARPIN-C4/ATPalphaS structure ?

In their rebuttal letter, the authors wrote 'ATP2 was indeed built based on the ATPalphaS conformation from crystal structure AC5_C1/AC2_C2 (PDB:1CJK)'. But in the method they state: '...based on the density elements corresponding to possible conformations of the bound MANT-GTP (M1 and M2) molecules present in the AC9_1250/GalphaS/M reconstruction'. It would be nice to clarify this point.

3/ Regarding AC9 conformation: what is the starting conformation of AC9 in the simulations? As I understand, it is AC9(+GalphaS) taken from AC9_1250+GalphaS+MANT-GTP from Figure 3e, which accommodates the M2 pose (Figure 3e). The M1 pose is seen in presence of Forskolin, in the AC9_1250+GalphaS+MANT-GTP+Forskolin structure. As shown in Figure C4, In AC9_1250+GalphaS+MANT-GTP+Forskolin, domain C1 is displaced further away from domain C2. This probably results in a larger binding pocket at the C1/C2 interface, able to accommodate the triphosphate moiety in a more vertical orientation (M1/A1). In these conditions, is it really surprising that A1 ATP is unstable when put in a A2/M2 binding pocket ?

4/ The authors conclude this part by saying that "location M1, albeit favourable for MANT-GTP, is somewhat unfavourable as an ATP-binding site".

I would rephrase this sentence. The term 'pose M1' seems more appropriate than 'location M1', since both ATP are localized in the same binding pocket in both simulations. Furthermore, ATPalphaS binds at this site in AC9+DARPIN-C4+ATPalphaS structure so the "somewhat unfavourable as an ATP-binding site" should be nuanced.

Response to reviewer comments

Reviewer #1 (Remarks to the Author):

1. This revised manuscript is improved from the original [redacted] submission; however, there remain a few substantive concerns. In the introduction, the authors cite controversy surrounding AC9 activation by forskolin, without including the (at least partially) enlightening study demonstrating that that mouse AC9 lacked key residues essential for forskolin activation (Yan et al Mol Pharm 1998).

RESPONSE: We are grateful to the reviewer for the positive comments on our revised manuscript. The stringent control and the critique by the reviewers helped us to improve our new revised manuscript. The reference to Yan et al is a match to the content of our manuscript, and we now cite this paper on page 2, line 75.

“Furthermore, the ability of forskolin to activate AC9 has been a controversial subject, with some studies indicating that the enzyme is insensitive to forskolin², some studies showing that mouse AC9 could be converted to forskolin sensitive cyclase by a Tyr1082Leu mutation¹⁶.”

2. As cited in my initial review, the rationale for including the molecular dynamic simulations was unclear. The authors wisely pursued and succeeded in obtaining additional structures which clarify the issue of ATP binding, which renders the molecular dynamic simulations even more extraneous in this revised manuscript. Without better justification, it remains unclear why they are included in this manuscript.

RESPONSE: In the original version of our manuscript we tried to use the MD simulations to better understand how ATP could bind, based on the experimental evidence. During the revision process we could determine the AC9-C4-ATP α S structure, showing that ATP α S binds the protein in a canonical pose. In light of this experimental finding the MD simulations are somewhat redundant. However, we believe that the observation derived from the MD simulations showing that ATP could not be stable in the ATP1 pose, which is similar to MANT-GTP1 pose, is still an interesting and valuable point. It is also consistent with our data and with the previous publications. We opt to keep the MD simulations in the manuscript as a piece of supplementary evidence.

We explain the rationale for the MD simulations that should provide the background for our attempts to understand how ATP may interact with the protein. We updated the text with the following description: (page 6, line 235):

“Although ATP binding and catalysis by ACs has been extensively investigated previously, the unusual binding pose of MANT-GTP observed in our structures raised a question whether an ATP molecule could also be accommodated in an alternative binding pose in the active site. To determine whether AC9 may have any preference for a specific nucleotide pose and to test that the M1 state may approximate a potential suitable nucleotide-bound conformation, we performed molecular dynamics (MD) simulations using the cytosolic portion of AC9-G α s model.”

3. Finally, as suggested by the reviews of the previous submission, the authors included discussion of the other mammalian adenylyl cyclase whose structures are known in both active and inactive states (soluble adenylyl cyclase; ADCY10). However, it is unclear why they exclusively focus on the structure of the ATP bound in the sAC structure bound to its inhibitor LRE1. There are structures of sAC bound to ATP analogs which do not include inhibitor, which would be a more relevant structure to use for comparison to an “active” AC9 structure.

RESPONSE: We used the sAC10-LRE1 example to show that ATP has been previously observed in non-canonical binding poses when bound to an adenylyl cyclase. When we compare the structure AC9 to sAC10, we indeed use the sAC10 apo (PDB: 4c1l) and bicarbonate bound (PDB: 4c1f) structure in Figure S16.

Reviewer #2 (Remarks to the Author):

In the revised manuscript, entitled ‘Structural Basis of adenylyl cyclase 9 activation’ by Qi et al, the authors made several modifications that improved the manuscript.

RESPONSE: We are grateful to the reviewer for this positive evaluation of our revised manuscript.

In many ways the real novelty of this study is the novel partial agonist, C4. However, in this revised manuscript the authors maintained the description and discussion of the alternative conformational states of AC9 stabilized by MANT-GTP, despite reviewers' sharp criticism of the relevance of MANT-GTP as a model for ATP binding.

The AC field has evolved significantly through the last decade or so and the biochemical and structural data describing MANT-GTP as a unique inhibitor are plentiful. There is a tendency for the authors to obfuscate the potential binding sites for ATP based on binding poses of the inhibitor MANT-GTP. As mentioned previously, the MANT moiety itself presents confounding effects on AC structure that ATP would never adopt.

RESPONSE: We acknowledge the criticisms of our original use of MANT-GTP that was raised by the reviewers, and we have worked hard to address this point. After extensive revisions, the following changes have been integrated into the manuscript:

- (i) We emphasized that MANT-GTP as an inhibitor, not a substrate analogue. We specifically state that MANT-GTP is used exclusively as a tool for structural studies, not as a substitute for ATP.
- (ii) We included additional experimental evidence with ATP α S, a new structure that is in line with the rest of our findings. MANT-GTP occupies the active site. The conformation of the protein in the MANT-GTP-bound state is similar to that in the ATP α S-bound state.
- (iii) It is our intention to not obfuscate and to not mix up the ATP and MANT-GTP, and we explicitly state this in the revised manuscript. The impression that we might be confusing the two states may have inadvertently resulted from the MD simulations, where we explore the possible ATP conformations. We feel that inclusion of the MD results does not undermine our story, but provides additional supporting evidence. This may be valuable especially in the absence of direct experimental structural data with ATP.

The fact that the triphosphates of MANT-GTP are not even poorly hydrolyzed would support this. In addition, the capacity of the MANT moiety to interfere with the presumptive Mg²⁺ coordination, and therefore the contributions of the highly conserved and highly important two aspartate residues in the C1 domain, argue that the fluorescent GTP molecule is not a good model for ATP. In all, the authors might be better off diminishing the emphasis on MANT-GTP as a model for alternative conformation binding states of ATP and enhance the description of the structure and mechanism of the novel partial agonist C4.

RESPONSE: We agree with the reviewer almost entirely: MANT-GTP is not a substrate for the cyclase, and it is a poor model for ATP binding – we are fully in agreement with all the issues listed by the reviewer. However, rather than studying ATP sub-states, we are using MANT-GTP in this case as a tool compound to stabilize the protein for structural studies and to evaluate the effects of different activators on the conformational states of the protein on a domain level (given the limited resolution of our reconstructions). We have included the points raised by the reviewer in our manuscript, to ensure that the reader understands this reasoning and is not misled into thinking that MANT-GTP mimics ATP, as follows (page 3, line 107):

“To characterize the conformation of AC9 in the absence of G protein α subunit, we determined the structure of bovine AC9 in complex with MANT-GTP (an inhibitor we used to stabilise AC9), referred to as AC9-M throughout the text below, using cryo-EM and single particle analysis at 4.9 Å resolution (Supplementary Fig. 1). MANT-GTP is not a substrate analogue of AC9, and way it is accommodated in the active site of the ACs is known to differ from that for ATP analogues. Our choice of MANT-GTP as a ligand for the active site of AC9 was based on two considerations: (i) we have used it successfully in our previous studies of membrane ACs, (ii) the overall conformation of the AC catalytic domains stabilized by MANT-GTP is similar to that stabilized by ATP analogues, such as ATP α S, based on X-ray crystallographic studies^{13,21}.”

Comments:

1. The real novelty of this study is the identification and structure of the C4 DARPin partial agonist. The authors include extra data on catalytic constants. Note that the authors should include the V_{max} values of AC9 in the presence of Galphas and Mn in their description of progress curves in the presence of C4. While

only 4-fold higher than basal activity, the V_{max} in the presence of Gs is approximately 24-fold higher than basal and 7.5 higher than C4 (assuming V_{max} with Gs ~ 900 nm/min/mg). The magnitude of the partially agonist activity needs to be put into perspective.

RESPONSE: We opted not to show / duplicate the published data - these experiments have been performed previously (Qi et al., Science, 2019; we included the reference to the original publication describing this data in the legend of the revised Figure S5). The corresponding data for the DARPin C4 can be readily compared to the V_{max} of AC9-G α_s , as requested by the reviewer.

2. The ‘in cellulo’ measurements (not in vivo) of C4 studies need to include data on whether AC9 expression was affected by C4 overexpression. Such studies would include measurements of plasma membrane expression and/or maximal adenylyl cyclase activity (Galphas with Mn²⁺). These studies could be hormone-stimulated.

RESPONSE: We have performed quantitative analysis of AC9 expression in the absence and in the presence of co-expressed DARPin C4, using three different techniques: (i) SDS PAGE & in-gel fluorescence, (ii) fluorescence size exclusion chromatography, (iii) confocal microscopy. The new data shows that DARPin C4 does not influence the expression of AC9, and we have included the results in the revised Figure S3.

3. The description of the mechanism of C4 activation compared to Galphas remains unsatisfying. Although both proteins engage the alpha2 and alpha3 helices of the C2 domain, only Galphas engages the C1 domain. In fact the entire N-terminal region of the C1 domain is disordered in the presence of the DARPin, highlighting its inability to full active AC9. It is understandable that the authors are trying to argue that activation of cyclase involves more than bringing the two domains together, which previous models based on the actions of forskolin have implied. However, the actions of Galphas, as a full agonist, engages both the C1 and C2 domains in a slightly different manner than forskolin, while C4, a partial agonist engages only the C2 domain. This is quite clear from the AC9-Galphas structures and those of the C1(AC5)-C2(AC2) cytosolic domains.

RESPONSE: In response to the previous reviewer comments, we moved away from describing forskolin action as merely bringing the two halves together. Forskolin appears to alter the conformation of the catalytic domain in more complex way, reorienting it for more efficient catalysis.

We have clarified further our interpretation of the differences between Gs- and DARPin C4-mediated activation, as follows (page 4, line 178):

“Furthermore, a loop in the C1a domain corresponding to residues I380-P384, appears to be flexible and could not be resolved in the AC9-C4-M map (Fig. 2f). However, this loop interacts with the G α_s protein, evident from the structure of the AC9-G α_s complex (Fig. 2g) as well as the available AC5_{c1}/AC2_{c2} crystal structures. G α_s as a full activator engages C1a and C2a domain simultaneously, whereas DARPin C4 as a partial activator only engages the C2a domain. This extensive interaction with the regions in both C1a and C2a domain of AC9 may contribute to higher potency of G α_s , compared to DARPin C4.”

4. The use of MANT-GTP still remains a little bit of an issue. Following the recommendations of the reviewers the authors removed state 3 from their models in substrate-bound mechanistic models. MANT-GTP is still an inhibitor and stabilizes a different conformational state than a substrate. The MANT moiety is not silent and plays a much larger role in AC9 binding than would ATP, or even cAMP, for that matter. This it is extremely difficult to understand the relevance of this state to ATP sub-states, particularly when the geometry of the 3'OH on the ribose ring is quite different with MANT-GTP compared to other non-hydrolyzable ATP analogues.

RESPONSE: We agree, MANT-GTP is not a substrate for the cyclase, and it is a poor model for ATP binding. However, we are using MANT-GTP in this case as a tool compound to stabilize the protein for structural studies and to evaluate the effects of different activators on the conformational states of the protein on a domain level. We try to address the possibility of accommodating ATP into the observed states using the MD simulations. In the We have included the cautionary points raised by the reviewers, to ensure that the reader understands this reasoning and is not mislead into thinking that MANT-GTP mimics ATP, as follows (page 3, line 107):

“To characterize the conformation of AC9 in the absence of G protein α subunit, we determined the structure of bovine AC9 in complex with MANT-GTP (an inhibitor we used to stabilise AC9), referred to as AC9-M

throughout the text below, using cryo-EM and single particle analysis at 4.9 Å resolution (Supplementary Fig. 1). MANT-GTP is not a substrate analogue of AC9, and way it is accommodated in the active site of the ACs is known to differ from that for ATP analogues. Our choice of MANT-GTP as a ligand for the active site of AC9 was based on two considerations: (i) we have used it successfully in our previous studies of membrane ACs, (ii) the overall conformation of the AC catalytic domains stabilized by MANT-GTP is similar to that stabilized by ATP analogues, such as ATP α S, based on X-ray crystallographic studies^{13,21}.”

We also explain the rationale for the MD simulations that should provide the background for our attempts to understand how ATP may interact with the protein (page 6, line 235):

“Although ATP binding and catalysis by ACs has been extensively investigated previously, the unusual binding pose of MANT-GTP observed in our structures raised a question whether an ATP molecule could also be accommodated in an alternative binding pose in the active site. To determine whether AC9 may have any preference for a specific nucleotide pose and to test that the M1 state may approximate a potential suitable nucleotide-bound conformation, we performed molecular dynamics (MD) simulations using the cytosolic portion of AC9-G α s model.”

5. The elongated density observed with the MANT-GTP extending into the forskolin binding site is interesting. This overlap would imply that forskolin binding may impair MANT-GTP binding. If the authors want to imply that the binding of MANT-GTP (and thus potentially GTP) with forskolin, then they should actually test this. This is easily testable since MANT-GTP is fluorescent. In this case fluorescence anisotropy measurements following MANT could be made in the presence and absence of forskolin (dose response curve of forskolin).

RESPONSE: Our data published in Qi et al., Science, 2019, argues that forskolin rather facilitates MANT-GTP binding, acting as an allosteric activator. In the presence of forskolin, the IC₅₀ of MANT-GTP shifts toward lower concentrations. This is the case for both the full-length AC9, and for AC9₁₂₅₀. This suggests that the forskolin- and MANT-GTP-bound state features a preferential pose M1 of MANT-GTP. We would refrain from performing further biophysical measurements, as they are redundant given the available data. The functional data for forskolin/MANT-GTP interactions (Qi et al., Science, 2019) serve as an excellent *in vitro* measure of the allosteric effect of forskolin. However, we have added a reference to these previous observations in our revised manuscript as follows (page 6, line 231):

“This new MANT-GTP pose is consistent with our previous observations of forskolin decreasing the IC₅₀ of MANT-GTP¹. The allosteric coupling between the two compounds is manifested by MANT-GTP moving from pose M2 to M1 upon forskolin binding.”

6. The authors appropriately raise the controversy regarding forskolin binding to AC9. As noted there do not appear to be any glaring side-chain incompatibilities in AC9 that would prevent forskolin binding, agreeing with literature reporting forskolin-mediated activation of AC9. The fact that the authors are able to delineate the structure of forskolin-bound AC9 validates this. The controversy, as implied by the authors, is likely based on how cyclase activity was measured. Most membrane-bound mammalian cyclases (but not all) display positively cooperative actions between Galphas and forskolin. The proximity of the two binding sites for these modulators makes structural sense for how this allostery could occur. Do the authors have data to measure this cooperativity exists to explain why forskolin, on its own, very poorly activates AC9 ?

RESPONSE: In our previously published study (Qi. et al, Science, 2019) we measured the forskolin dose-response in the absence and presence of G α s. We found that AC9 is weakly activated by forskolin. However, AC9 is activated by forskolin in the presence of G α s, suggesting the cooperativity between G α s and forskolin. We clarified this point as follows (page 2, line 75):

“Furthermore, the ability of forskolin to activate AC9 has been a controversial subject, with some studies indicating that the enzyme is insensitive to forskolin², some studies showing that mouse AC9 could be converted to forskolin sensitive cyclase by a Tyr1082Leu mutation¹⁶. Recently, the cryo-EM structure of AC9₁₂₅₀-G α s bound to MANT-GTP and forskolin, combined with biochemical studies, confirmed that AC9 can be activated by forskolin binding to its canonical allosteric site in the presence of G α s^{1,17}”.

It is worth pointing out that until now we do not understand why forskolin fails to efficiently activate AC9 alone. This remains an enigma, and we hope that in the future this can be addressed. A simple explanation is that AC9

in the absence of G protein has an allosteric binding pocket incompatible with high affinity binding of forskolin. G α s binding (and C4 binding, as we have now ascertained, as detailed in the continued response below) induces a conformation that is more conducive to forskolin binding.

Forskolin dose-response measurements in the absence and presence of Galphas would be appropriate. Such data would easily address the apparent controversy regarding forskolin-insensitivity of this isoform. Similar experiments should also be made using the partial agonist C4.

RESPONSE: Such experiments have been performed for the AC9-G α s complex (Qi et al., Science, 2019), and the data showed that forskolin activates AC9 in the presence of G α s – in the absence of the G protein the activation is very weak. Following the reviewer's recommendation, we have performed similar dose-response measurements for forskolin activation in the presence of DARPin C4. We found that forskolin could activate AC9-C4 complex (EC₅₀ of ~130 μ M), but not AC9 alone, suggesting DARPin C4 is inducing a conformational change that allows the protein to accommodate forskolin. The EC₅₀ value is in the range that was measured for AC9-G α s complex. The results are included in the revised Figure S5C.

7. The mystery still remains why Galphas binding appears to stabilize the auto-inhibitory state in the cryoEM structure. The authors cite work on phosphorylation as a possible contributor however they need to describe, at least to speculate how phosphorylation may affect the structure of the C-terminal region of C2. This could be depiction of the putative phosphorylation sites relative to the C-term in the auto-inhibited state.

RESPONSE: As previously reported by Palvolgyi, A. et al, Cell Signal, 2018, the C-terminal region of AC9 autoinhibits the catalytic activity of the protein. This was validated by our first structure of AC9 in an occluded state (Qi et al., Science, 2019). We have extended the discussion to reflect this, as described below (page 7, line 280):

“This autoregulation may be phosphorylation-dependent, as suggested by a recent mutagenesis-based study of AC9 autoinhibition by its C-terminus²⁴.”

Minor comments:

In line 202 the authors refer to ATP analogue ApCpp. Are they referring to AMP-Cpp ? They should refer this.

RESPONSE: We kept ApCpp to be consistent with the original publication, and we now included “AMP-Cpp” in brackets to ensure the ligand identity is clear to the readers more accustomed to the alternative name of the compound (page5, line 210).

Reviewer #4 (Remarks to the Author):

The manuscript entitled "Structural basis of adenylyl cyclase 9 activation" reports several full-length structures of the membrane protein AC9, acquired by cryo-EM, in in complex with natural or a novel synthetic activator proteins and nucleotide analogues. The structural data are accompanied by appropriate biochemical characterization of the synthetic activator, in vitro and in vivo enzymatic assays and in vivo microscopy, in order to progress in the understanding of AC activation. This is an impressive amount of work and it is highly appreciated having this type of study for full-length membrane proteins.

RESPONSE: We thank the reviewer for the kind words and for the positive evaluation of our work.

I have evaluated only the part related to MD simulation. In this part, the authors simulated the soluble domains of AC9 in complex with the natural protein activator Galphas and ATP bound in two different conformations:

- Conformation A1, derived from the MANT-GTP conformation M1 seen in the AC9_1250/Galphas/MANT-GTP/Forskolin structure (previously published),
- Conformation A2, derived from the MANT-GTP in conformation M2 seen in the AC9/Galphas/MANT-GTP structure presented here.

They observed that only conformation A2 is stable, supporting the M2 conformation of MANT-GTP. My main recommendation would be to better articulate this analysis with the structural data presented for AC9+DARPIN-C4+ATP α S. Since there are structural data with ATP bound, it is not completely clear what MD simulations bring to the study.

1. Regarding the motivations, the authors sought to: 1) ‘To determine whether AC9 may have any preference for a specific nucleotide pose’ and 2) ‘to ascertain that the M2 state may represent a stable nucleotide-bound conformation’. I would think that Figure 3 already addresses both points: MANT-GTP can adopt two different conformations, depending on the presence of forskolin (or other ligand in this site), as shown in Figure 3e and 3f. In the absence of forskolin, MANT-GTP occupies a part of the forskolin binding site in the M2 conformation, stably enough to observe density at this site. Although the stability of A2 confirms that the proposed model of MANT-GTP M2 conformation is reasonable, the MD analysis indeed raises further questions.

RESPONSE: We are grateful to the reviewer for pointing out the insufficiently described rationale for using the MD simulations. In the revised manuscript we have clarified our intentions more precisely. Our original goal was to try and rationalize the presence of the unusual density of MANT-GTP in the pocket (which previously has not been observed in any of the available structure of adenylyl cyclases), as well as to understand whether ATP molecules could interact with the active site in a similar manner to MANT-GTP.

As the reviewer surmised, the M2 state appears to be reasonable and both the experimental data (MANT-GTP-bound cryo-EM structures of AC9 reported) and the MD simulations (using ATP instead of MANT-GTP, in the ATP2 pose) suggest that M2 state of MANT-GTP is reasonable. While this is not a central point of our study, we felt that using the MD data here was appropriate. Consistent with this and as expected, our ATP α S-bound AC9 structure also shows the ATP2-like conformation.

The result is consistent with the previously determined X-ray structures of the adenylyl cyclases. As reviewer 2 points out, MANT-GTP is a unique inhibitor that can be oriented in the binding site in poses that are different from those available for ATP (i.e., M1 and M2 poses, dependent on the presence of forskolin).

2. Regarding the ATP conformations used in the simulations: What is the agreement between the M1 pose and the density of MANT-GTP seen in AC9₁₂₅₀+GalphaS+MANT-GTP ? Looking at Figures 3e and 3f/13a, it seems rather limited, which makes it a questionable starting point for a simulation with ATP in the A1 pose.

RESPONSE: M1 pose is indeed not matching to the density, because it corresponds to AC9₁₂₅₀-G α s bound to MANT-GTP and FSK (Figure 3F). The M2 pose is corresponding to AC9₁₂₅₀-G α s bound to MANT-GTP (Figure 3E). ATP1 as a starting point for MD simulations corresponds to this pose M1, but effectively serves as a control that confirms that this is not a stable pose for ATP.

Is the A2 conformation similar to ATP α S in the AC9/DARPIN-C4/ATP α S structure ? In their rebuttal letter, the authors wrote ‘ATP2 was indeed built based on the ATP α S conformation from crystal structure AC5_{C1}/AC2_{C2} (PDB:1CJK)’. But in the method they state: ‘...based on the density elements corresponding to possible conformations of the bound MANT-GTP (M1 and M2) molecules present in the AC9₁₂₅₀/GalphaS/M reconstruction’. It would be nice to clarify this point.

RESPONSE: The A2 conformation is similar to ATP α S in the AC9-DARPinC4-ATP α S structure. We inadvertently used an older version of the Methods section and overlooked this point after the first revision. The revised manuscript reads as follows (Methods, page 4):

“Two ATP conformations (A1 and A2) were manually built in COOT, based on the density elements corresponding to MANT-GTP in the structure AC9₁₂₅₀-G α s-MF (PDB: 6R4O) and in the crystal structure AC5_{C1}-AC2_{C2}-G α s-ATP α S (PDB: 1CJK) and AC5_{C1}-AC2_{C2}-G α s-ATP (PDB:3C16)”

3. Regarding AC9 conformation: what is the starting conformation of AC9 in the simulations? As I understand, it is AC9(+GalphaS) taken from AC9₁₂₅₀+GalphaS+MANT-GTP from Figure 3e, which accommodates the M2 pose (Figure 3e). The M1 pose is seen in presence of Forskolin, in the AC9₁₂₅₀+GalphaS+MANT-GTP+Forskolin structure. As shown in Figure C4, In AC9₁₂₅₀+GalphaS+MANT-GTP+Forskolin, domain C1 is displaced further away from domain C2. This probably results in a larger binding pocket at the C1/C2 interface, able to accommodate the triphosphate

moiety in a more vertical orientation (M1/A1). In these conditions, is it really surprising that A1 ATP is unstable when put in a A2/M2 binding pocket ?

RESPONSE: The model used for MD simulation is that of AC9₁₂₅₀+G α s+MANT-GTP. We actually placed ATP1 into the M1 binding pocket, and similarly ATP2 was also placed into the M2 binding pocket. The interesting result was that despite the presumably better match of ATP1 to M1, this pose of ATP was unstable. The ATP2 was a better pose for the nucleotide, consistent with all available knowledge about ATP-cyclase interactions, as mentioned by Reviewer 2.

4. The authors conclude this part by saying that “location M1, albeit favourable for MANT-GTP, is somewhat unfavourable as an ATP-binding site”. I would rephrase this sentence. The term ‘pose M1’ seems more appropriate than ‘location M1’, since both ATP are localized in the same binding pocket in both simulations. Furthermore, ATPalphaS binds at this site in AC9+DARPIN-C4+ATPalphaS structure so the “somewhat unfavourable as an ATP-binding site” should be nuanced.

RESPONSE: We have rephrased this sentence as follows (page 6, line 243):

“Taken together, the simulations performed suggest that although pose M1 is favourable for MANT-GTP in the presence of forskolin, a similar pose for ATP would be unfavourable.”

REVIEWER COMMENTS

Reviewer #1 (Remarks to the Author):

The authors have satisfactorily addressed my previous critiques. In its current form, the work is significant and makes a substantive contribution to the field.

Reviewer #2 (Remarks to the Author):

The revised manuscript by Qi et al represents a significant improvement over the previous submission. The authors address most, but not all this reviewer's queries. The authors have provided additional citations and referred to these studies to support the models proposed in current manuscript. The authors have also framed the rationale for using molecular dynamics simulations based on the MANT-GTP structures a little more clearly. The significance of the MANT-GTP conformations that are not supported by ATP binding still raise questions regarding the MANT-GTP structure's physiological relevance. Not to say that the simulations, as casted in revised manuscript, are not useful. In fact, it could be argued that if an enzyme is capable of supporting such conformations and nucleotide binding then simulating ATP binding to determine the relevance of the conformations, such as presented here, are worth testing.

Comments:

The authors state the 3-fold enhancement of the DARPin C4 over basal cyclase activity but only referred to the Gs-stimulated activity as being higher. As mentioned in the previous critique, the authors should include the fold-enhancement in activity compared to Gs so that readers can put the 'agonist' efficacy of C4 into perspective. The authors have the Vmax values of Gs-stimulated activity with full length and the truncated AC9. From the figure it appears the Vmax with Galphas of the non-occluded form is approximately 750 nmol/min/mg and perhaps 400 nmol/min/mg for the occluded (full length). This would represent a ~ 20-fold enhancement in the non-occluded state and 12-fold enhancement in the occluded state.

The authors raise an interesting mechanism depicted in Figure 5. As illustrated, the authors suggest that in the presence of substrate ATP, Galphas does not stabilize the occluded state. With cellular ATP levels in the 1-10 millimolar range then what is the relevance of a nucleotide-free, occluded state? Might this serve as a feedback to sense if cellular ATP concentrations dip below the Km for ATP of cyclase (~25 uM)? Since it's presented in the figure it should be discussed. Adenylyl cyclase assay conditions used in the assay (100 uM ATP) may not be high enough to overcome this occlusion.

The discussion of why AC9 is less sensitive to forskolin could be a little more in depth and may in fact be related to basal activity. It could be that forskolin is capable of binding and stabilizing active conformations of other cyclase isoforms because they have higher basal activity. Thus AC9, which happens to have low basal activity may not be able to support forskolin binding in its apo state. In the presence of C4 or Galphas, in contrast, forskolin appears to bind relatively well ($EC_{50} \sim 120 \mu M$). Albeit, the affinity of forskolin may be slightly lower owing to the Y1082L substitution, but this would make it less sensitive but not insensitive. The authors' data here and in the previous structure argues for a less sensitive but not insensitive.

Minor Comments:

Throughout the manuscript the authors refer to the induction of states. It might be preferable to address the effects of C4, forskolin and Galphas as 'stabilizing specific conformations', rather than 'inducing' conformations.

The figure 4 is a nice comparison of the structures but the authors should consider illustrating where Galphas binds ($\alpha 2$ - $\alpha 3$ of C2a), with respect to C4. This could simply be represented by 3-10-helix of switch II of Galphas.

In Figure 5 the authors should label the $\alpha 4$ helix so that readers do not have look at the details of the legend.

Reviewer #4 (Remarks to the Author):

The authors responded satisfactorily to my questions about the conformations of ATP and AC9 used in their MD simulations. The fact that both simulations are done with AC9 in the conformation from AC9_1250+Gs+MANT-GTP without Forskolin -the one that displays MANT-GTP in conformation M2- should be included in the Method part, and not only in the response to the reviewers.

In view of this precision, I am still dubious about the way they justify the necessity to run MD simulations, notably the simulation with ATP in A1.

The justify as: "[...] the unusual binding pose of MANT-GTP observed in our structures raised a question whether an ATP molecule could also be accommodated in an alternative binding pose in the active site.

To determine whether AC9 may have any preference for a specific nucleotide pose and to test that the M1 state may approximate a potential suitable nucleotide-bound conformation”.

The structure of AC9-C4-A (Figure 3b) answers the first point (‘whether an ATP molecule could also be accommodated in an alternative binding pose’), showing the ATP in the A2 conformation. The simulation with ATP/A2 bound to AC9 in place of MANT-GTP in the AC9_1250+Gs+M conformation gives additional support.

The utility of the simulation with ATP A1 is less clear, as well as the conclusions drawn from this simulation.

What is the motivation ‘to test that the M1 state may approximate a potential suitable nucleotide-bound conformation,’ given the AC9-C4-A structure (Figure 3b) showing ATP bound in the A2 conformation (Figure 3b)?

In my previous report, I asked the authors if the instability of ATP A1 was surprising, given the fact that AC9 has different conformations to accommodate M2 (AC9_1250+Gs+MANT-GTP Figure 4c, RMSD 3.15) and M1 (AC9_1250+Gs+MANT-GTP+Forskolin Figure 4d, RMSD 5.7).

Now that it is clear that AC9 was in the M2-compatible conformations in both simulations, my question remains open.

As a consequence, I am not convinced by the conclusions drawn from the simulation with ATP A1: “Taken together, the simulations performed suggest that although pose M1 is favourable for MANT-GTP in the presence of forskolin, a similar pose for ATP would be unfavourable”.

To support this affirmation, a simulation should be conducted with AC9 in the conformation AC9_1250+Gs+MANT-GTP+Forskolin, with Forskolin.

Without this simulation, it is difficult to interpret the ATP A1 stability, which could be the mere result of the different conformation of AC9. The role of Forskolin could be tested with simulations with and without Forskolin.

Without such additional simulations, I recommend putting less emphasis on the ATP A1 simulation in manuscript.

RESPONSE TO REVIEWER COMMENTS

Reviewer #1 (Remarks to the Author):

The authors have satisfactorily addressed my previous critiques. In its current form, the work is significant and makes a substantive contribution to the field.

Reviewer #2 (Remarks to the Author):

The revised manuscript by Qi et al represents a significant improvement over the previous submission. The authors address most, but not all this reviewer's queries. The authors have provided additional citations and referred to these studies to support the models proposed in current manuscript. The authors have also framed the rationale for using molecular dynamics simulations based on the MANT-GTP structures a little more clearly. The significance of the MANT-GTP conformations that are not supported by ATP binding still raise questions regarding the MANT-GTP structure's physiological relevance. Not to say that the simulations, as casted in revised manuscript, are not useful. In fact, it could be argued that if an enzyme is capable of supporting such conformations and nucleotide binding then simulating ATP binding to determine the relevance of the conformations, such as presented here, are worth testing.

Comments:

1. The authors state the 3-fold enhancement of the DARPin C4 over basal cyclase activity but only referred to the Gs-stimulated activity as being higher. As mentioned in the previous critique, the authors should include the fold-enhancement in activity compared to Gs so that readers can put the 'agonist' efficacy of C4 into perspective. The authors have the Vmax values of Gs-stimulated activity with full length and the truncated AC9. From the figure it appears the Vmax with Galphas of the non-occluded form is approximately 750 nmol/min/mg and perhaps 400 nmol/min/mg for the occluded (full length). This would represent a ~20-fold enhancement in the non-occluded state and 12-fold enhancement in the occluded state.

RESPONSE: We thank the reviewer for this suggestion, and we include the following statement on page 4, line 133, addressing the fold-increase in AC9 activity:

"In comparison, G α s potentially activated the full-length AC9 with a ~15-fold increase for the full-length protein and a ~24-fold increase for AC9₁₂₅₀ (i.e., in the absence of the C2b domain) (Fig. 1d)."

2. The authors raise an interesting mechanism depicted in Figure 5. As illustrated, the authors suggest that in the presence of substrate ATP, Galphas does not stabilize the occluded state. With cellular ATP levels in the 1-10 millimolar range then what is the relevance of a nucleotide-free, occluded state? Might this serve as a feedback to sense if cellular ATP concentrations dip below the Km for ATP of cyclase (~25 μ M)?

RESPONSE: This is an intriguing suggestion. The occluded state may indeed represent a feedback loop for ATP sensing. Our cautious interpretation of this state is that it is involved in decreasing the AC9 activity in response to a yet unknown stimulus, as part of a complex regulatory mechanism. As pointed in the Palvolgyi, A. et al. 2018, the phosphorylated C2b domain of AC9 is supporting the occluded state. So it is possible that this state represents an auto-inhibited state of the AC under conditions where cAMP production needs to be limited.

We have modified the legend for the Figure 5, to explain more clearly what this occluded state likely represents. The reader should not get the wrong impression about our interpretation of this state, and the previous legend was not clear – we have corrected this (page 14, line 427):

“.. The stop-arrows indicate inability of the nucleotide and/or forskolin to bind to the AC in the occluded state.”

Since it's presented in the figure it should be discussed. Adenylyl cyclase assay conditions used in the assay (100 uM ATP) may not be high enough to overcome this occlusion.

RESPONSE: With respect to overcoming the occluded state, as reviewer correctly points out, we have no hope to achieve this task with increased nucleotide concentration: our original occluded state was determined with AC9-Gas in the presence of 0.5 mM MANT-GTP and 0.5 mM forskolin. Only by removing the C-terminus of AC9 could we obtain a MANT-GTP and forskolin bound state. However, only a fraction of total particles in the sample resulted in a high resolution reconstruction – so it is likely that the sample of AC9-Gas contains a mixed population, as we can see the enzymatic activity. This activity is further increased by removal of the C-terminus (the C2b domain), as we have shown in Qi et al., Science, 2019.

Our choice of ATP concentration (100 uM) in the assays was based on this value exceeding the K_m for ATP. The goal was not to overcome the occluded state (an ungrateful task, as we pointed out above), but to determine the AC activity under a set condition and to estimate the relative effects upon changes of the conditions (e.g., in the absence of G protein, in the presence of C4, etc).

3. The discussion of why AC9 is less sensitive to forskolin could be a little more in depth and may in fact be related to basal activity. It could be that forskolin is capable of binding and stabilizing active conformations of other cyclase isoforms because they have higher basal activity. Thus AC9, which happens to have low basal activity may not be able to support forskolin binding in its apo state. In the presence of C4 or Galphas, in contrast, forskolin appears to bind relatively well (EC_{50} ~120 uM). Albeit, the affinity of forskolin may be slightly lower owing to the Y1082L substitution, but this would make it less sensitive but not insensitive. The authors' data here and in the previous structure argues for a less sensitive but not insensitive.

RESPONSE: This is an important point - our data indeed show that AC9 is less sensitive but not completely insensitive to forskolin. We have added a statement reflecting this idea in the text (page 2, line 81):

“.., further suggesting that AC9 is not completely insensitive to forskolin.”

Minor Comments:

1. Throughout the manuscript the authors refer to the induction of states. It might be preferable to address the effects of C4, forskolin and Galphas as ‘stabilizing specific conformations’, rather than ‘inducing’ conformations.

RESPONSE: As suggested by the reviewer, we now use “stabilization”, rather than “induction” in many of the cases throughout the manuscript. We keep “induction” in several cases, as it appears reasonable in that specific context.

2. The figure 4 is a nice comparison of the structures but the authors should consider illustrating where Galphas binds (alpha2-alpha3 of C2a), with respect to C4. This could simply be represented by 3-10-helix of switch II of Galphas.

RESPONSE: To ensure that this is well illustrated, we have now the labeled $\alpha 2$ and $\alpha 3$ in the corresponding Figure 4 to make the $G\alpha s$ binding site easily recognizable.

3. In Figure 5 the authors should label the a4 helix so that readers do not have look at the details of the legend.

Response: We have now added the $\alpha 4$ helix label in Figure 5 to make the presentation more clear.

Reviewer #4 (Remarks to the Author):

1. The authors responded satisfactorily to my questions about the conformations of ATP and AC9 used in their MD simulations. The fact that both simulations are done with AC9 in the conformation from AC9_1250+Gs+MANT-GTP without Forskolin -the one that displays MANT-GTP in conformation M2- should be included in the Method part, and not only in the response to the reviewers.

RESPONSE: As suggested by reviewer, we have introduced the following comment to the Methods section (page 4 of Supplementary information):

“The simulations were performed using the AC9₁₂₅₀-G α s-M model (in the absence of forskolin).”

In view of this precision, I am still dubious about the way they justify the necessity to run MD simulations, notably the simulation with ATP in A1.

The justify as: “ [...] the unusual binding pose of MANT-GTP observed in our structures raised a question whether an ATP molecule could also be accommodated in an alternative binding pose in the active site. To determine whether AC9 may have any preference for a specific nucleotide pose and to test that the M1 state may approximate a potential suitable nucleotide-bound conformation”.

The structure of AC9-C4-A (Figure 3b) answers the first point (‘whether an ATP molecule could also be accommodated in an alternative binding pose’), showing the ATP in the A2 conformation. The simulation with ATP/A2 bound to AC9 in place of MANT-GTP in the AC9_1250+Gs+M conformation gives additional support.

The utility of the simulation with ATP A1 is less clear, as well as the conclusions drawn from this simulation.

What is the motivation ‘to test that the M1 state may approximate a potential suitable nucleotide-bound conformation,’ given the AC9-C4-A structure (Figure 3b) showing ATP bound in the A2 conformation (Figure 3b)?

RESPONSE: The MD simulations of ATP at A1 state confirmed the point that ATP did not have an alternative binding pose, because ATP could not be stabilized in the A1 state, only in the A2 state. We also want to point out that at the beginning of the project, we could not determine the structure of AC9-C4-A, and thus we tried to use MD simulations to understand the possible behavior of ATP in A1 or A2 state. During the revision process we were able to determine the structure with ATP α S (AC9-C4-A), clearly indicating the pose of the nucleotide. The structure of AC9-C4-A was compatible with the MD simulation, confirming that ATP is stable in the A2 state. We believe that including the ATP1 simulation in the revised manuscript is still logically justified, and we can argue that it is sufficiently interesting to perform such a simulation, even if it is effectively a “negative control”.

We feel that the motivation to perform this MD analysis was correct at the time when we did it, and inclusion of the statement about our motivation now does not compromise the story. We modified the statement to reduce the chances of misinterpretation as follows (page 6, line 240):

“Although ATP binding and catalysis by ACs has been extensively investigated previously, the unusual binding pose of MANT-GTP observed in our structures raised a question whether an ATP molecule could also be accommodated in an alternative binding pose in the active site. To determine whether AC9 may have any preference for a specific nucleotide pose, we performed molecular dynamics (MD) simulations using the cytosolic portion of AC9-G α s model.”

In my previous report, I asked the authors if the instability of ATP A1 was surprising, given the fact that AC9 has different conformations to accommodate M2 (AC9_1250+Gs+MANT-GTP Figure 4c, RMSD 3.15) and M1 (AC9_1250+Gs+MANT-GTP+Forskolin Figure 4d, RMSD 5.7).

Now that it is clear that AC9 was in the M2-compatible conformations in both simulations, my question remains open.

RESPONSE: To ensure that we do not overlook this point this time, we confirm that this was not surprising. The previous crystal structures and our own results point to the instability of ATP in the A1 state. We used the MD simulation as a tool to confirm this.

As a consequence, I am not convinced by the conclusions drawn from the simulation with ATP A1: “Taken together, the simulations performed suggest that although pose M1 is favourable for MANT-GTP in the presence of forskolin, a similar pose for ATP would be unfavourable”.

To support this affirmation, a simulation should be conducted with AC9 in the conformation AC9_1250+Gs+MANT-GTP+Forskolin, with Forskolin. Without this simulation, it is difficult to interpret the ATP A1 stability, which could be the mere result of the different conformation of AC9. The role of Forskolin could be tested with simulations with and without Forskolin.

Without such additional simulations, I recommend putting less emphasis on the ATP A1 simulation in manuscript.

RESPONSE: As the reviewer suggested, to make our interpretation more clear, we put less emphasis on the ATP1 simulation in our manuscript and show that ATP only could be stable in the ATP2 state (page 6, line 240).

“Although ATP binding and catalysis by ACs has been extensively investigated previously, the unusual binding pose of MANT-GTP observed in our structures raised a question whether an ATP molecule could also be accommodated in an alternative binding pose in the active site. To determine whether AC9 may have any preference for a specific nucleotide pose, we performed molecular dynamics (MD) simulations using the cytosolic portion of AC9-G α s model. We substituted the MANT-GTP molecules in M1 and M2 poses with the molecules of ATP placed in the corresponding poses ATP1 and ATP2 (Supplementary Fig. 14a-d). During the MD simulations molecules placed at ATP2 remained bound at their starting locations (the RMSDs of the nucleotide atoms were around 2 Å; Supplementary Fig. 14d; Movie 2). Molecules at ATP1 shifted from their initial location (Supplementary Fig. 14c; Movie 1). Taken together, the simulations performed suggest that although pose M1 is favourable for MANT-GTP, a similar pose for ATP would be unfavourable, which is consistent with the previous studies. It is likely that MANT-GTP, an AC inhibitor, is stabilized in the corresponding M1 conformation through additional π - π stacking interactions of the MANT group (e.g., with the W1188 residue in AC9).”